# SENSEI: Semantic Exploration Guided by Foundation Models to Learn Versatile World Models

**Cansu Sancaktar** [* 1 2] **Christian Gumbsch** [* 1 3 4] **Andrii Zadaianchuk** [1 5] **Pavel Kolev** [1] **Georg Martius** [1 2]

## Abstract

Exploration is a cornerstone of reinforcement learning (RL). Intrinsic motivation attempts to decouple exploration from external, task-based rewards. However, established approaches to intrinsic motivation that follow general principles such as information gain, often only uncover low-level interactions. In contrast, children's play suggests that they engage in meaningful high-level behavior by imitating or interacting with their caregivers. Recent work has focused on using foundation models to inject these semantic biases into exploration. However, these methods often rely on unrealistic assumptions, such as language-embedded environments or access to high-level actions. We propose SEmaNtically Sensible ExploratIon (SENSEI), a framework to equip model-based RL agents with an intrinsic motivation for semantically meaningful behavior. SENSEI distills a reward signal of interestingness from Vision Language Model (VLM) annotations, enabling an agent to predict these rewards through a world model. Using model-based RL, SENSEI trains an exploration policy that jointly maximizes semantic rewards and uncertainty. We show that in both robotic and video game-like simulations SENSEI discovers a variety of meaningful behaviors from image observations and low-level actions. SENSEI provides a general tool for learning from foundation model feedback, a crucial research direction, as VLMs become more powerful.[1]

## 1. Introduction

Achieving intrinsically-motivated learning in artificial agents has been a long-standing dream, making it possible to decouple agents' learning from an experimenter manually crafting and setting up tasks. Thus, the goal in intrinsically-motivated reinforcement learning (RL) is for agents to explore their environment efficiently and autonomously, constituting a free play phase akin to children's curious play. Various intrinsic reward definitions have been proposed in the literature, such as aiming for state space coverage (Bellemare et al., 2016; Tang et al., 2017; Burda et al., 2019), novelty or retrospective surprise (Pathak et al., 2017; Schmidhuber, 1991), and information gain of a world model (Pathak et al., 2019; Sekar et al., 2020; Sancaktar et al., 2022). However, when an agent starts learning from scratch, there is one fundamental problem: just because something is novel does not necessarily mean that it contains useful or generalizable information for any sensible task (Dubey & Griffiths, 2017).

Imagine a robot facing a desk with several objects. The robot could explore by trying to move through the entire manipulable space or hitting the desk at various speeds. In contrast, human common sense would primarily focus on interacting with the objects or drawer of the desk since potential task distributions likely revolve around those entities.

Agents exploring their environment with intrinsic motivations suffer from a chicken-or-egg problem: *how do you know something is interesting before you have tried it and experienced interesting consequences?* This is a bottleneck for the types of behavior that an agent can unlock during free play. We argue that incorporating human priors into exploration could alleviate this roadblock. Similar points have been raised for children's play. During the first years of life, children are surrounded by their caregivers who ideally encourage and reinforce them while they explore their environment. Philosopher and psychologist Karl Groos has stipulated that there is "a strong drive in children to observe the activities of their elders and incorporate those activities into their play" (Gray, 2017; Groos & Baldwin, 1901).

A potential solution in the age of Large Language Models (LLMs), is to utilize language as a cultural-transmitter to inject "human notions of interestingness" (Zhang et al.,

---

*Equal contribution [1]Autonomous Learning, University of Tübingen [2]Empirical Inference, Max Planck Institute for Intelligent Systems [3]Neuro-Cognitive Modeling, University of Tübingen [4]Cognitive and Clinical Neuroscience, TU Dresden [5]VISLab, University of Amsterdam. Correspondence to: Cansu Sancaktar <cansu.sancaktar@tue.mpg.de>, Christian Gumbsch <chris@gumbsch.de>.

*Proceedings of the 42nd International Conference on Machine Learning*, Vancouver, Canada. PMLR 267, 2025. Copyright 2025 by the author(s).

[1]Project website with videos and code: https://sites.google.com/view/sensei-paper

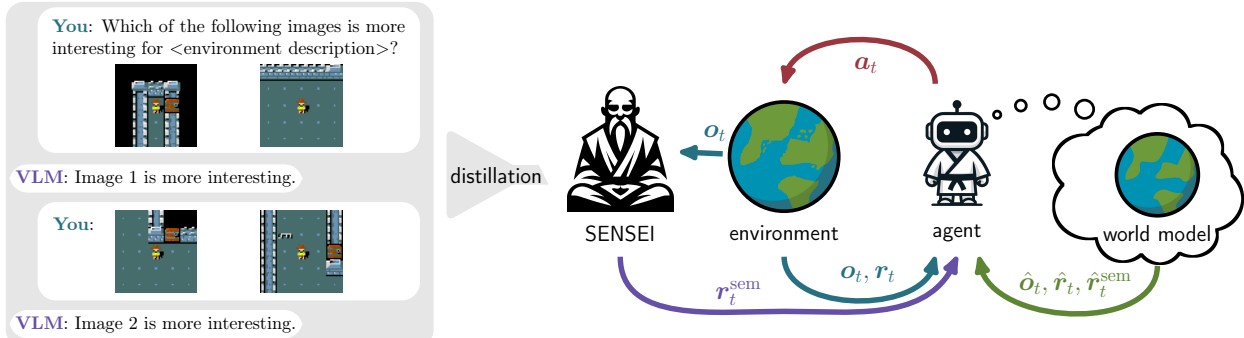

*Figure 1.* **SENSEI overview**: **(a)** During pre-training we prompt a VLM to compare observations (e.g. images) from an environment with respect to their interestingness. We distill this ranking into a reward function (SENSEI) for exploration. **(b)** An exploring agent not only receives observations ($o_t$) and rewards ($r_t$) from interactions with the environment, but also a semantic exploration reward ($r_t^{\text{sem}}$) from SENSEI. **(c)** The agent learns a world model from its experience to judge the interestingness ($\hat{r}_t^{\text{sem}}$) of states without querying SENSEI.

2023a) into RL agents' exploration. LLMs are trained on an immense amount of data produced mostly by humans. Thus, their responses are likely to mirror human preferences. However, the most prominent works in this domain assume (1) a language-grounded environment (Zhang et al., 2023b; Du et al., 2023; Klissarov et al., 2023), (2) the availability of an offline dataset with exhaustive state-space coverage (Klissarov et al., 2023), or (3) access to high-level actions (Zhang et al., 2023a; Du et al., 2023). These assumptions are detached from the reality of embodied agents, e.g. in robotics, which don't come with perfect state or event captioners, pre-existing offline datasets nor with robust, high-level actions. Furthermore, none of these approaches learn an internal model of "interestingness." Thus, they rely on the LLM, or a distilled module, to continuously guide their exploration.

In this work, we propose SEmaNtically Sensible ExploratIon (SENSEI), a framework for Vision Language Model (VLM) guided exploration for model-based RL agents, illustrated in Fig. 1. SENSEI starts with a short description of the environment and a dataset of observations (e.g. images) collected through self-supervised exploration. A VLM is prompted to compare the observations pairwise with respect to their interestingness and the resulting ranking is distilled into a reward function. When the agent explores its environment, it receives semantically-grounded exploration rewards from SENSEI. It learns to predict this exploration signal through its learned world model, corresponding to an internal model of "interestingness". The agent improves its exploration strategy by aiming for states for which it predicts a high interestingness and then branching out to uncertain situations. Our main contributions are as follows:

- We propose SENSEI, a framework for foundation model-guided exploration with world models.

- We show that SENSEI can explore rich, semantically meaningful behaviors with few prerequisites.

- We demonstrate that the versatile world models learned

through SENSEI enable fast learning of downstream tasks.

## 2. Method

We consider the setup of an agent interacting with a Partially Observable Markov Decision Process. At each time $t$, the agent performs an action $a_t \in \mathcal{A}$ and receives an observation $o_t \in \mathcal{O}$, composed of an image and potentially additional information. We assume that there exist one or more tasks in the environment with corresponding rewards $r_t^{\text{task}} \in \mathbb{R}$. However, during task-free exploration, the agent should select its behavior agnostic to task rewards.

We assume that SENSEI starts with a dataset $\mathcal{D}^{\text{init}} \subset \mathcal{O}$, collected from self-supervised exploration with information gain as intrinsic reward (Sekar et al., 2020), thus not relying on a pre-existing expert dataset. SENSEI has access to a pretrained VLM and is provided with a short description of the environment, either from a human expert or generated by the VLM, based on some observations from $\mathcal{D}^{\text{init}}$. *Prior to* task-free exploration, SENSEI distills a semantic exploration reward function from VLM annotations (Sec. 2.1). *During* exploration, SENSEI learns a world model (Sec. 2.2) and optimizes an exploration policy through model-based RL (Sec. 2.3).

### 2.1. Reward function distillation: MOTIFate SENSEI

Prior to task-free exploration, SENSEI needs to distill a semantically grounded intrinsic reward function $R_\psi$ with learnable parameters $\psi$ based on the preferences of a pretrained VLM. While the overall framework of SENSEI is agnostic to the exact distillation method, we chose to use a vision-based extension of MOTIF (Klissarov et al., 2023; illustrated in Fig. 2a), which we refer to as VLM-MOTIF.[2]

---

[2]Original MOTIF (Klissarov et al., 2023) assumes an environment where events are captioned in natural language. Thus, they can use LLMs to annotate captions of observations.

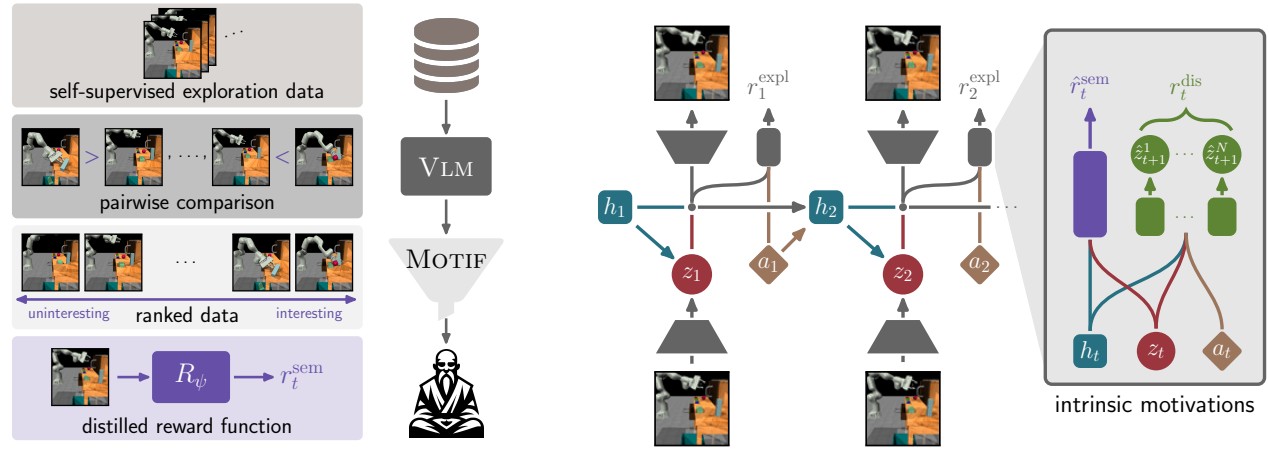

(a) reward function distillation  (b) world model

*Figure 2.* **Intrinsic rewards in SENSEI: (a)** Prior to exploration, we prompt GPT-4 to compare images with respect to the interestingness for a certain environment. From the resulting ranking we distill a reward function $R_\psi$ using VLM-MOTIF. **(b)** Later, an agent learns an RSSM world model from exploration. From each state the agent predicts two intrinsic rewards, i.e., the distilled semantic rewards $r_t^{\text{sem}}$ and uncertainty-based rewards $r_t^{\text{dis}}$.

MOTIF consists of two phases. In the first phase of **dataset annotation**, the pretrained foundation model is used to compare pairs of observations, creating a dataset of preferences. For this, we prompt the VLM with an environment description and pairs of observations from $\mathcal{D}^{\text{init}}$, asking the VLM which image it considers to be more interesting. The annotation function is given by the VLM : $\mathcal{O} \times \mathcal{O} \to \mathcal{Y}$, where $\mathcal{O}$ is the space of observations, and $\mathcal{Y} = \{1, 2, \emptyset\}$ is a space of choices for the first, second or none of the observations. In the **reward training** phase, a reward function is derived from the VLM preferences using standard techniques from preference-based RL (Wirth et al., 2017). A cross-entropy loss function is minimized on the dataset of preference pairs to learn a semantically grounded reward model $R_\psi : \mathcal{O} \to \mathbb{R}$. We use the final semantic reward function $R_\psi$ whenever the agent interacts with its environment: the agent not only receives an observation $\boldsymbol{o}_t$ and reward $\boldsymbol{r}_t$ after executing an action $\boldsymbol{a}_t$, but also receives a semantically-grounded exploration reward $r_t^{\text{sem}} \leftarrow R_\psi(\boldsymbol{o}_t)$ (see Fig. 1, center).

## 2.2. World model: Let your SENSEI dream

We assume a model-based setting, i.e., the agent learns a world model from its interactions. Following DreamerV3 (Hafner et al., 2023), we implement the world model as a Recurrent State Space Model (RSSM) (Hafner et al., 2019b). The RSSM with learnable parameters $\phi$ is computed by

$$\text{Posterior:} \quad \boldsymbol{z}_t \sim q_\phi(\boldsymbol{z}_t \mid \boldsymbol{h}_t, \boldsymbol{o}_t) \tag{1}$$

$$\text{Dynamics:} \quad \boldsymbol{h}_{t+1} = f_\phi(\boldsymbol{a}_t, \boldsymbol{h}_t, \boldsymbol{z}_t) \tag{2}$$

$$\text{Prior:} \quad \hat{\boldsymbol{z}}_{t+1} \sim p_\phi(\hat{\boldsymbol{z}}_{t+1} \mid \boldsymbol{h}_{t+1}) \tag{3}$$

In short, the RSSM encodes all interactions through two latent states, a stochastic state $\boldsymbol{z}_t$ and a deterministic memory

$\boldsymbol{h}_t$. At each time $t$, the RSSM samples a new stochastic state $\boldsymbol{z}_t$ from a posterior distribution $q_\phi$ computed from the current deterministic state $\boldsymbol{h}_t$ and new observation $\boldsymbol{o}_t$ (Eq. 1). The RSSM updates its deterministic memory $\boldsymbol{h}_{t+1}$ based on the action $\boldsymbol{a}_t$ and previous latent states (Eq. 2). Then, the model predicts the next stochastic state $\hat{\boldsymbol{z}}_{t+1}$ (Eq. 3). Once the new observation $\boldsymbol{o}_{t+1}$ is received, the next posterior $q_\phi$ is computed and the process is repeated.

Besides encoding dynamics within its latent state, the RSSM is also trained to reconstruct external quantities $y_t$ from its latent state via output heads $o_\phi$:

$$\text{Output heads:} \quad \hat{y}_t \sim o_\phi(\hat{y}_t \mid \boldsymbol{h}_t, \boldsymbol{z}_t) \tag{4}$$

with $y_t \in \{\boldsymbol{o}_t, c_t, r_t, r_t^{\text{sem}}\}$. The RSSM of DreamerV3 (Hafner et al., 2023) reconstructs observations $\boldsymbol{o}_t$, episode continuations $c_t$, and rewards $r_t$. For SENSEI, we additionally predict the semantic exploration reward $r_t^{\text{sem}}$. The world model is trained end-to-end to jointly optimize the evidence lower bound.

Thus, our world model learns to predict semantic interestingness $\hat{r}_t^{\text{sem}}$ of states (see Fig. 1, right). We could base exploration exclusively on this signal. However, we (1) expect to face many local optima when optimizing for this signal and (2) we do not want to only explore a fixed set of behaviors, but ensure that the agent goes for interesting and yet novel states. To overcome this limitation, Klissarov et al. (2023) post-process $r_t^{\text{sem}}$ and normalize it by episodic event message counts. As we do not assume ground-truth countable event captions, we instead **combine our semantic reward signal with epistemic uncertainty**, a quantity that was shown to be an effective objective for model-based exploration (Sekar et al., 2020; Pathak et al., 2017; Sancaktar et al., 2022). Following Plan2Explore (Sekar et al.,

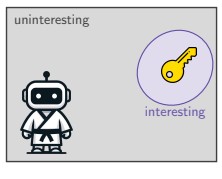

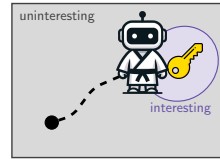

(a) **go:** $\hat{r}_t^{\text{sem}} < Q_k$       (b) **explore:** $\hat{r}_t^{\text{sem}} \geq Q_k$

*Figure 3.* **Switching exploration behavior in** SENSEI: **(a)** When the agent is in an uninteresting state it mainly strives to maximize "interestingness" ($r_t^{\text{sem}}$), e.g. by going to a key. **(b)** When in an interesting state, the agent more strongly attempts to increase uncertainty ($r_t^{\text{dis}}$) by trying new actions, e.g. picking up the key.

2020), we train an ensemble of $N$ models with weights $\{\theta^1, \ldots, \theta^N\}$ to predict the next stochastic latent states with

$$\text{Ensemble predictor:} \quad \hat{z}_t^n \sim g_{\theta^n}(\hat{z}_t^n \mid h_t, z_t, a_t). \quad (5)$$

We quantify epistemic uncertainty as ensemble disagreement $r_t^{\text{dis}}$ by computing the variance over the ensemble predictions averaged over latent state dimensions $J$:

$$r_t^{\text{dis}} = \frac{1}{J} \sum_{j=1}^{J} \text{Var}(\hat{z}_{j,t}^n), \quad (6)$$

Thus, the model learns to predict two intrinsic rewards ($\hat{r}_t^{\text{sem}}, r_t^{\text{dis}}$) for a state-action-pair (Fig. 2b).

### 2.3. Exploration policy: Go and Explore with SENSEI

We could use a weighted sum of the two intrinsic reward signals, e.g. $r_t^{\text{sem}} + \beta r_t^{\text{dis}}$, as the overall reward $r_t^{\text{expl}}$ for optimizing an exploration policy. However, ideally the weighting of the two signals should depend on the situation. In uninteresting states, we want the agent to mostly pursue interestingness (via $r_t^{\text{sem}}$). However, once the agent has found an interesting state, we would like the agent to branch out and discover new behavior (via $r_t^{\text{dis}}$). This follows the principle of Go-Explore (Ecoffet et al., 2021), where the agent should first **go** to a subgoal and **explore** from there (illustrated in Fig. 3). We implement this using an adaptive threshold parameter $\beta \in \{\beta^{\text{go}}, \beta^{\text{explore}}\}$, with $\beta^{\text{explore}} > \beta^{\text{go}}$, whose value depends on the following switching criteria:

$$r_t^{\text{expl}} = \hat{r}_t^{\text{sem}} + \begin{cases} \beta^{\text{explore}} r_t^{\text{dis}}, & \text{if } \hat{r}_t^{\text{sem}} \geq Q_k(\hat{r}^{\text{sem}}); \\ \beta^{\text{go}} r_t^{\text{dis}}, & \text{otherwise.} \end{cases} \quad (7)$$

Here $Q_k$ denotes the $k^{\text{th}}$ quantile of $\hat{r}^{\text{sem}}$, which we estimate through a moving average. Thus, until a certain level of $\hat{r}^{\text{sem}}$ is reached, the exploration reward mainly aims to maximize interestingness. After exceeding this threshold, exploration more strongly favors uncertainty-maximizing behaviors. As soon as the agent enters a less interesting state with $\hat{r}^{\text{sem}} < Q_k$, SENSEI switches back to focusing

on semantic interestingness. The two trade-off factors $\beta^{\text{go}}$ and $\beta^{\text{explore}}$, as well as the quantile $k$ are hyperparameters. More details on this adaptation and hyperparameters can be found in Suppl. B. We learn an exploration policy based on $r_t^{\text{expl}}$ using DreamerV3 (Hafner et al., 2023).

## 3. Related work

**Intrinsic rewards** are applied either to facilitate exploration in tasks where direct rewards are sparse or in a task-agnostic setting where they help collect diverse data. Many different reward signals have been proposed as exploration rewards (Baldassarre & Mirolli, 2013), such as prediction error (Schmidhuber, 1991; Pathak et al., 2017; Kim et al., 2020), Bayesian surprise (Storck et al., 1995; Blaes et al., 2019; Paolo et al., 2021), learning progress (Schmidhuber, 1991; Colas et al., 2019; Blaes et al., 2019), empowerment (Klyubin et al., 2005; Mohamed & Jimenez Rezende, 2015), metrics for state-space coverage (Bellemare et al., 2016; Tang et al., 2017; Burda et al., 2019) and regularity (Sancaktar et al., 2024). While effective for low-dimensional observations, such objectives are challenging to apply for high-dimensional image observations. Here, alternatives are employing low-dimensional goal spaces (Colas et al., 2019; OpenAI et al., 2021; Nair et al., 2018; Pong et al., 2019; Zadaianchuk et al., 2021; Mendonca et al., 2021) or learning latent world models (Hafner et al., 2019a; 2023; Gumbsch et al., 2024) that can be employed for model-based exploration (Pathak et al., 2019; Sekar et al., 2020). In particular, Plan2Explore (Sekar et al., 2020) uses ensemble disagreement of latent space dynamics predictions as an intrinsic reward. While this is a very general strategy for exploration, this could be limited in more challenging environments where semantically meaningful or goal-directed behavior (Spelke, 1990) is needed for efficient exploration.

**Exploration with foundation models:** Recent improvements of in-context learning of LLMs open additional ways to explore using human bias of interestingness during exploration (Klissarov et al., 2023; Du et al., 2023; Zhang et al., 2023a) and skill learning (Colas et al., 2020; 2023; Zhang et al., 2023b). MOTIF (Klissarov et al., 2023) leverages LLMs to derive intrinsic rewards by comparing pairs of event captions, demonstrating its efficacy in the complex game of NetHack (Küttler et al., 2020). Similarly, ELLM (Du et al., 2023) uses LLMs to guide RL agents towards goals that are meaningful, based on the agent's current state represented as text. Furthermore, OMNI (Zhang et al., 2023a) introduces a novel method to prioritize tasks using LLMs. Thereby, OMNI focuses on tasks that are not only learnable but also generally interesting. LAMP (Adeniji et al., 2023) proposes to use VLMs for reward modulation by first generating a set of potential tasks with an LLM and then generating task-based rewards using VLMs.

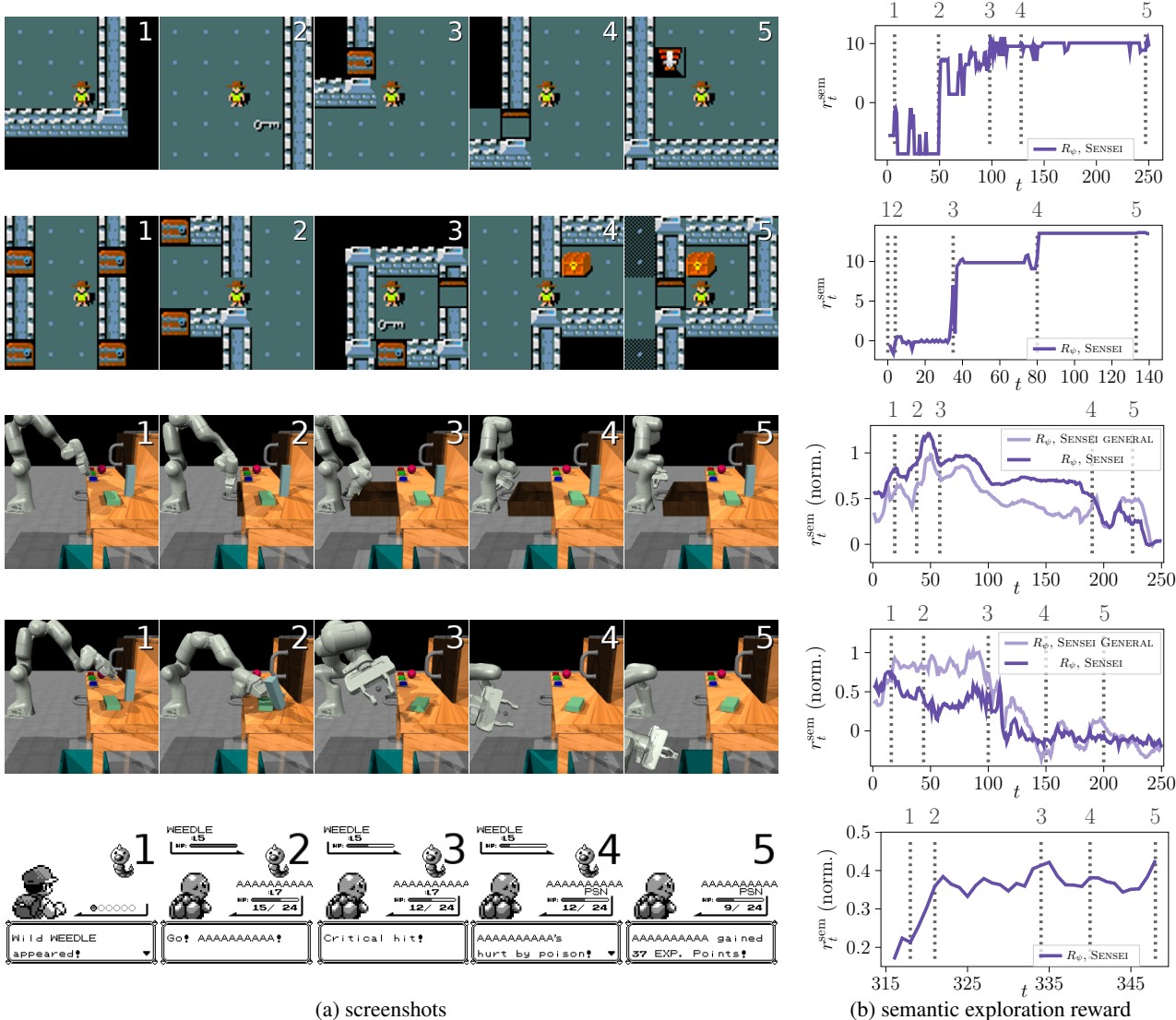

|(a) screenshots|(b) semantic exploration reward|

*Figure 4.* **Semantic exploration rewards for example trajectories**: From top to bottom we show example trajectories for MiniHack `KeyRoom` and `KeyChest` (see Fig. 10 for map view), Robodesk, and Pokémon Red. We showcase rewards from VLM-MOTIF distilled from GPT-4 annotations. The reward trajectories peak at the "interesting" moments of exploration, such as opening a drawer in Robodesk, picking up the key in MiniHack, or landing a critical hit in Pokémon. For Robodesk we show reward trajectories for both SENSEI and a version of SENSEI with a more general, zero pre-knowledge prompting strategy (SENSEI GENERAL, see Sec. 4.2.2).

**Reward-shaping through VLMs:** Most works that rely on VLMs as reward sources try to solve the reward specification problem in RL. In these works, a task is assumed to be described as a language caption (Cui et al., 2022; Rocamonde et al., 2023; Baumli et al., 2023; Adeniji et al., 2023), as a goal image (Cui et al., 2022), or as a video demonstration (Sontakke et al., 2023). In particular, RL-VLM-F (Wang et al., 2024) uses a very similar setup to SENSEI. Pairs of images from initial rollouts are compared using a VLM to distill a reward function via MOTIF (Klissarov et al., 2023). However, we assume a model-based setup and do not explicitly prompt the task and distill an environment-specific but general exploration reward.

## 4. Results

Our experiments set out to empirically evaluate the following questions:

1. Does the distilled reward function $R_\psi$ from VLM annotations encourage interesting behavior?
2. Can SENSEI discover semantically meaningful behavior during task-free exploration?
3. Is the world model learned via exploration suitable for later learning to efficiently solve downstream tasks?
4. Can SENSEI be combined with extrinsic rewards to solve tasks that require substantial exploration?

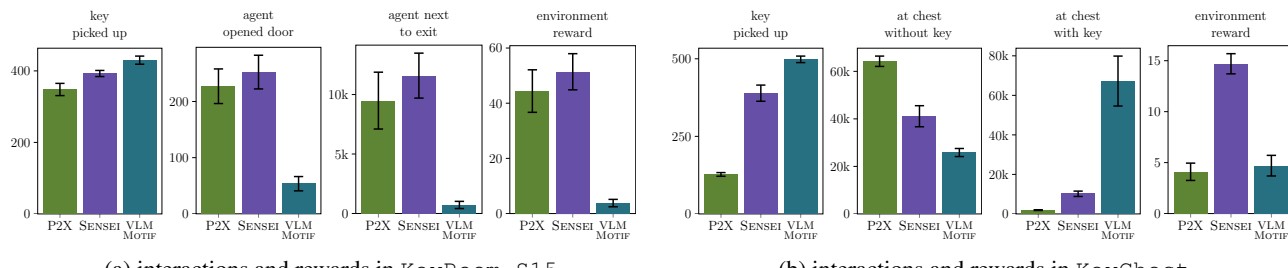

(a) interactions and rewards in `KeyRoom-S15`          (b) interactions and rewards in `KeyChest`

*Figure 5.* **Interactions in MiniHack**: We plot the mean number of interactions with task-relevant objects and the environment reward (unknown to the agents) collected by SENSEI, Plan2Explore (P2X) and pure VLM-MOTIF (SENSEI with no information gain, i.e. $\beta = 0$) for `KeyRoom-S15` (**a**) and `KeyChest` (**b**). Error bars show the standard error (10 seeds).

We answer these questions by (1) illustrating that the semantic rewards obtained from VLM-MOTIF reflect interesting events in the environment, (2) quantitatively showing that SENSEI leads to more interaction-rich behavior during task-free exploration, (3) employing the learned world models to successfully train task-based policies and (4) combining SENSEI's exploration strategy with extrinsic rewards to tackle a challenging environment that cannot be efficiently explored using rewards alone. We use three fundamentally different types of environments:

**MiniHack** (Samvelyan et al., 2021) is a sandbox to design RL tasks based on NetHack (Küttler et al., 2020). In MiniHack, an agent needs to navigate dungeons by meaningfully interacting with its environment, e.g. open a door with a key. We tested two tasks: fetching a key in a large room to unlock a smaller room with an exit (`KeyRoom-S15`) or fetching a key to open a chest in a maze of rooms (`KeyChest`). MiniHack uses discrete actions. As observations we use pixel-based, egocentric views around the agent and a binary flag indicating key pick-ups (details in Suppl. C.2).

**Robodesk** (Kannan et al., 2021) is a multi-task RL benchmark in which a simulated robotic arm can interact with various objects on a desk, including buttons, two types of blocks, a ball, a sliding cabinet, a drawer, and a bin. For different objects, there exist different tasks, e.g. `open_drawer`. Robodesk uses pixel-based observations and continuous actions. In order to deal with occlusions, we use images from two camera angles for VLM annotations but only one camera angle as input to our agents (details in Suppl. C.1).

**Pokémon Red** is a Game Boy role-playing game where players control a trainer exploring a world of collectible creatures called Pokémon. Exploration is essential for progress, as players must navigate a vast, interconnected world and master a semantic battle system (e.g. Water beats Rock) to defeat Gym leaders and become Pokémon Champion. Our implementation (Whidden, 2023; Suarez, 2024) use the raw game screen as pixel-based observations and discrete actions corresponding to Game Boy button presses (see Suppl. C.3).

In all environments, we collect the initial dataset $\mathcal{D}^{\text{init}}$ with Plan2Explore (Sekar et al., 2020), the current state-of-the art in exploration with pixel-based observations. We collect data from 500k steps in MiniHack and Pokémon Red and 1M steps in Robodesk. For data annotation, we use GPT-4 (details in Suppl. D).

### 4.1. Reward function of SENSEI

In Fig. 4 we illustrate how the distilled VLM-MOTIF reward function $R_\psi$ assigns semantic rewards $r_t^{\text{sem}}$ for exemplary sequences. In MiniHack, $r_t^{\text{sem}}$ clearly jumps for significant events. Frames 2 & 3 in `KeyRoom-S15` and `KeyChest` respectively, are right before the key is picked up. Later, $r_t^{\text{sem}}$ increases further once the agent is at the door or chest with a key (Frame 3 in `KeyRoom-S15` and Frames 4&5 in `KeyChest`). For Robodesk, we see that as the robot is interacting with objects, $r_t^{\text{sem}}$ increases, e.g., when opening the drawer or pushing the blocks. For Pokémon Red, $r_t^{\text{sem}}$ rises while winning a battle, with surges for inflicting damage and drops for setbacks such as getting poisoned. More examples of Robodesk are shown in Suppl. Fig. 17 and examples of Pokémon Red in Suppl. Fig. 26.

### 4.2. Task-free exploration

#### 4.2.1. MINIHACK

We quantify the interactions uncovered by SENSEI during task-free exploration in two tasks of MiniHack. For task-relevant events, the mean number of interactions are plotted in Fig. 5. SENSEI focuses more on semantically interesting interactions compared to Plan2Explore, e.g. picking up a key, opening a locked door, or finding the chest with a key. As a result, SENSEI completes both tasks more frequently than Plan2Explore during task-free exploration, as evident by the higher number of collected rewards. We believe this indicates that SENSEI is well suited for initial task-free exploration in these environments, enabling the discovery of state-space regions crucial for solving downstream tasks.

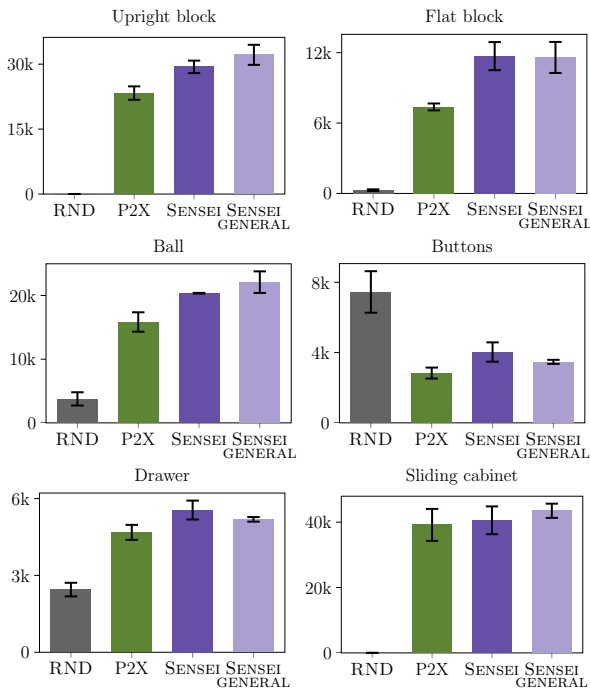

*Figure 6.* **Interactions in Robodesk**: We plot the mean number of object interactions during 1M steps of exploration for SENSEI (environment description provided by us), a more general variant of SENSEI with a VLM-generated environment description (SENSEI GENERAL), Plan2Explore (P2X), and Random Network Distillation (RND). Error bars show the standard deviation (3 seeds).

**Is information gain crucial for SENSEI?** We show results for exploration with pure semantic reward $r_t^{\text{sem}}$, corresponding to SENSEI without an information gain reward $r_t^{\text{dis}}$ ($\beta = 0$). In this VLM-MOTIF ablation, we emphasize the crucial role of the information gain objective. Optimizing only for the semantic reward $r_t^{\text{sem}}$ can cause the agent to get stuck in local optima and hinder further exploration. For example in KeyRoom, the agent with VLM-MOTIF often picks up the key. However, it fails to explore the room well enough after key pick-ups to find and open the door and reach the exit, as reflected in the interaction metrics in Fig. 5. We observe a similar scenario for KeyChest: although the pure VLM-MOTIF agent reaches the chest often after having picked up the key, it collects substantially less rewards than SENSEI. For the episode to end, the agent needs to use the key to open the chest. The VLM-MOTIF agent, however, simply hovers around the chest. As being at the chest with a key is an "interesting" state and opening a chest immediately terminates the episode, there is no real incentive for the agent to explore chest openings. This ablation shows the importance of **combining novelty and usefulness** in order to continually **push the frontier of experience**.

### 4.2.2. ROBODESK

Next, we analyze exploration in the challenging visual control suite of Robodesk. Here we compare 1M steps of exploration in SENSEI with Plan2Explore and Random Network Distillation (RND, Burda et al., 2019), a strong model-free exploration approach that uses prediction errors of random image embeddings as intrinsic rewards to maximize state space coverage. Fig. 6 plots the mean number of object interactions during exploration for the three methods. On average, SENSEI interacts more with most available objects than the baselines. As a result, in a majority of tasks SENSEI receives more task rewards during exploration than Plan2Explore or RND (shown in Suppl. E.3). Qualitatively, we observe that Plan2Explore mostly performs arm stretches[3], whereas RND mostly moves the arm around in the center of the screen, mostly hitting buttons, as they are also centered on the table, and occasionally hitting objects.

Thus, our semantic exploration scheme leads to more object interactions than uncertainty-based exploration, even in a low-level motor control robotic environment.

**Is an environment description by a human expert necessary for SENSEI?** In the previous SENSEI experiments, we provided a small environment description in the prompts for the VLM annotations. We investigate whether SENSEI relies on this external description in Robodesk, and compare against a version of SENSEI using a more general, zero-knowledge prompting strategy (SENSEI GENERAL). SENSEI GENERAL first prompts the VLM for an environment description given an image of the environment and uses the generated answer as context to annotate the dataset of preferences (details in Suppl. D.2). As shown in Fig. 6, SENSEI GENERAL interacts roughly as often with the relevant objects as SENSEI, outperforming both Plan2Explore and RND in terms of overall number of object interactions. Thus, injecting external environment knowledge to the prompts is not necessary and this step can be fully automated. This further cements the generality of our approach.

**Ablations** We perform ablations to see (1) how noisy annotations from VLMs affect SENSEI compared to an oracle annotator, (2) how much the behavior richness of the initial dataset affects SENSEI's performance (Suppl. Fig. 14), and (3) ablate our Go-explore switching strategy. We observe that as VLMs get better, there is indeed more to gain from SENSEI, and richer exploration data helps SENSEI bootstrap faster. See Suppl. E.2 for more information. We further showcase the robustness of our Go-Explore switching strategy in terms of hyperparameter sensitivity compared to a variant of SENSEI where the semantic and disagreement rewards get fixed weights (Suppl. E.6).

---

[3]Interestingly, this can still lead to solving tasks. For example, stretching the arm against the sliding cabinet can close it, and stretching the arm toward a block can push it off the table.

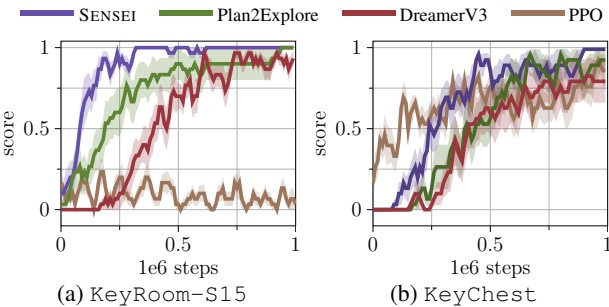

(a) KeyRoom-S15  (b) KeyChest

*Figure 7.* **Downstream task performance in MiniHack**: Mean episode scores for (**a**) KeyRoom-S15 and (**b**) KeyChest, using world models learned during task-free exploration with SENSEI and Plan2Explore (P2X). We also compare learning from scratch with DreamerV3 and PPO. Shaded areas show standard error (10 seeds); curves are smoothed with a window size of 3.

### 4.3. Fast downstream task learning

We hypothesize that world models learned from richer exploration would enable model-based RL agents to quickly learn to solve new downstream tasks. We investigate this in MiniHack by running DreamerV3 (Hafner et al., 2023) using the previously explored world models to learn a novel task-based policy. To this end, we initialize DreamerV3 with the pre-trained world models from the initial 500K steps of exploration (see Sec. 4.2). We compare world models from task-free exploration with either SENSEI or Plan2Explore. Additionally, we compare running DreamerV3 and training Proximal Policy Optimization (PPO, Schulman et al. 2017), a state-of-the-art model-free baseline, from scratch.

Figure 7 shows the performance of task-based policies over training. A previously explored world model from SENSEI allows the agent to learn to solve the task faster than all other baselines. Compared to Plan2Explore, SENSEI allocates more resources to explore the relevant dynamics in the environment, e.g. opening the chest more, resulting in well-suited world model for policy optimization. Unlike the clear improvements of SENSEI, task-free exploration with Plan2Explore does not outperform learning a task policy from scratch with DreamerV3 consistently across environments. In KeyRoom, the model-free baseline PPO takes more than 20M steps to consistently solve the task (full PPO curves in Supp. Fig. 13). Thus, in this task SENSEI outperforms PPO by roughly two orders of magnitude. This shows the improved sample efficiency of our approach: combining foundation model-guided exploration and model-based RL. In KeyChest, the model-free baseline PPO shows the first successes in the tasks early during training, but on average takes longer to learn to reliably solve the task.

In a supplementary experiment (Suppl. E.7), we analyze fast downstream task learning also on representative Robodesk tasks, and demonstrate more sample-efficient policy learning compared to exploration with Plan2Explore.

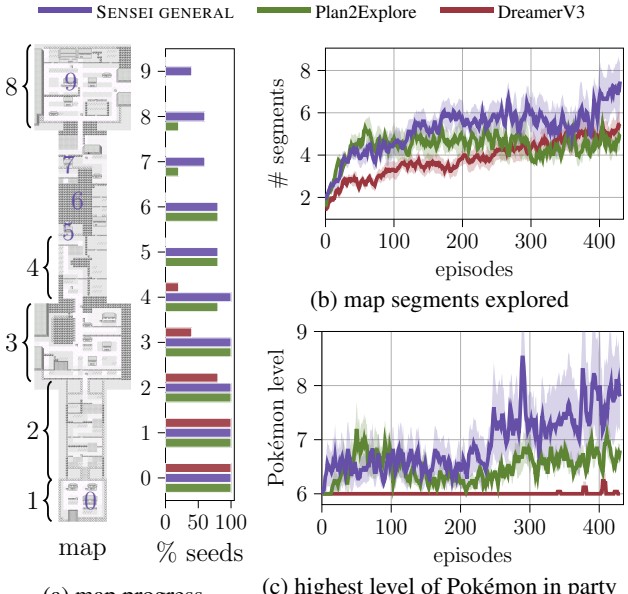

(b) map segments explored

(a) map progress  (c) highest level of Pokémon in party

*Figure 8.* **Task-based exploration in Pokémon Red** comparing SENSEI GENERAL to Plan2Explore and DreamerV3 for 750k steps. We partition the overall game map into unique map segments for different routes, towns, or buildings (details and full map in Suppl. C.3). We sequentially numbered segments that need to be traversed from game start (0) to the first Gym (9) and plot the percentage of random seeds that reach each segment (**a**). Temporal exploration trends are visualized by plotting the mean number of unique map segments visited (**b**) and the highest level of the agent's Pokémon (**c**) over episodes, smoothed with a moving average (window size 5). Shaded areas indicate standard error (5 seeds).

### 4.4. Task-based exploration

Some environments are so complex that meaningful task progress is only possible through effective exploration. We investigate such task-based exploration in Pokémon Red. We compare 750k steps of exploration with SENSEI GENERAL (using $r_t + r_t^{\text{expl}}$) to Plan2Explore (using $r_t + \beta r_t^{\text{dis}}$) and DreamerV3 (using only $r_t$). Thus, all agents receive extrinsic task rewards $r_t$ during exploration, but only Plan2Explore and SENSEI also utilize intrinsic signals.

Progress in the game requires navigating a vast world to reach Pokémon Gyms, as well as assembling and training a strong team to defeat Gym Leaders. To evaluate **spatial exploration**, we partition the game world into distinct map segments, corresponding to towns, routes, forests, or buildings (details in Suppl. C.3). Over the course of exploration, SENSEI consistently discovers new segments, outperforming baselines in terms of total map coverage (Fig. 8b). To assess whether this exploration is goal-directed, Figure 8a labels the specific segments required to progress from the game start (segment 0) to the first Gym (segment 9), and plots the segments reached per method (high-resolution map

in Fig. 12). Only SENSEI reaches the first Gym, demonstrating superior exploration aligned with the game's objectives.

To assess **battle-related progress**, we track the levels of the agent's Pokémon (Fig. 8c), which serve as a proxy for battle experience and overall strength. Dreamer fails to sufficiently explore the battle system and does not manage to train its Pokémon. Its highest-level Pokémon remains at the same level as at the start of the episode. In contrast, SENSEI begins leveling up its Pokémon early during exploration, and from episode 100 onward, it consistently achieves higher levels than Plan2Explore. From episode 390 onward, SENSEI, on average, obtains twice as many level-ups per episode as Plan2Explore, indicating greater battle success and a higher potential for future encounters.

Together, these results highlight SENSEI's ability to perform meaningful, goal-directed exploration in rich, open-ended environments. We provide a more detailed analysis in Suppl. E.8.

**Second generation of annotations**  With more exploration, and due to the vastness of the world in Pokémon Red, SENSEI increasingly enters regions outside the distribution of its annotation data ($\mathcal{D}^{\text{init}}$). This leads to degraded semantic rewards. In Suppl. E.9, we show how this limitation can be addressed by refining the reward function through a second round of VLM annotations on SENSEI-collected data. When continuing the SENSEI run used for annotation for 200 additional episodes, now using the updated reward function, the agent is able to defeat the first Gym and obtain the Boulder Badge. This marks a critical milestone, highlighting the strong potential of iterative semantic reward refinement to unlock meaningful progress in complex environments.

## 5. Discussion

We have introduced SENSEI, a framework for guiding the exploration of model-based agents through foundation models. SENSEI bootstraps a model of interestingness from previously generated play data. On this dataset, SENSEI prompts a VLM to compare images with respect to their interestingness and distills a semantic reward function. SENSEI learns an exploration policy via model-based RL using two sources of intrinsic rewards: (1) trying to reach states with high semantic interestingness and (2) branching out from these states to maximize epistemic uncertainty. We show that in the video game environments of MiniHack and Pokémon Red and a robotic simulation, this strategy leads to more meaningful interactions, e.g. opening a chest with a key or manipulating objects on a desk.

**Internal model of interestingness**  Unlike prior work of foundation model-guided exploration (Klissarov et al., 2023;

Wang et al., 2024), SENSEI learns an internal model of interestingness. This is a sensible design choice when working with world models (as detailed in Suppl. A.3), enabling SENSEI to predict semantic rewards also while imagining states during policy training. We demonstrate that this can lead to significantly faster learning, since both VLM guidance as well as model-based RL improve sample efficiency.

**Limitations**  SENSEI benefits from fully-observable observations, e.g. images that capture all relevant aspects of the environment. The VLM annotations degrade when dealing with occlusions. In Robodesk we mitigate this using multiple camera angles. In future work this could be remedied further by annotating videos to better convey temporal or partially-observable information.

**Future work**  One promising direction is to systematically investigate iterative refinement of the semantic reward function. In Suppl. E.9, we show that incorporating SENSEI-collected data into a second round of annotations reduces out-of-distribution errors for states not present in the initial annotation set. We believe this iterative process can unlock increasingly complex behaviors with each generation. Another avenue is to explore SENSEI in photorealistic or real-world environments. Photorealism of observations are likely to help VLM annotations because a large portion of VLMs' training data comes from real world photos or videos. Thus, SENSEI is likely to scale well to these settings.

## Acknowledgements

The authors thank Sebastian Blaes and Onno Eberhard for helpful discussions. The authors thank the International Max Planck Research School for Intelligent Systems (IMPRS-IS) for supporting Cansu Sancaktar and Christian Gumbsch. Georg Martius is a member of the Machine Learning Cluster of Excellence, EXC number 2064/1 – Project number 390727645. We acknowledge the financial support from the German Federal Ministry of Education and Research (BMBF) through the Tübingen AI Center (FKZ: 01IS18039B). This work was supported by the Volkswagen Stiftung (No 98 571).

## Impact Statement

This work introduces a framework for semantically meaningful exploration in reinforcement learning (RL), guided by intrinsic rewards distilled from vision-language models (VLMs). The approach enables agents to efficiently discover useful, high-level behaviors without relying on task-based rewards. RL often suffers from computational inefficiencies due to extensive trial-and-error processes, but effective exploration strategies can alleviate this by guiding agents towards more purposeful behaviors. In real-world set-

tings, exploration poses additional challenges due to safety concerns, as aimless interactions can lead to damage or unsafe situations. By emphasizing semantically meaningful exploration, our approach offers a step toward more energy-efficient and potentially safer exploration. We have identified no significant ethical concerns beyond standard considerations for responsibly deploying autonomous learning agents.

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

# Supplementary Material for:
# SENSEI: Semantic Exploration Guided by Foundation Models to Learn Versatile World Models

## A. SENSEI: Implementation Details

We provide code with VLM-MOTIF checkpoints at https://github.com/martius-lab/sensei.

### A.1. World model

**RSSM**  We base our RSSM implementation on DreamerV3 (Hafner et al., 2023). For MiniHack we use the small model size setting with roughly 18M parameters ($h_t$ dimensions: 512, CNN multiplier: 32, dense hidden units: 512, MLP layers: 2). For the more complicated Robodesk environment, we use the medium model size with around 37M parameters ($h_t$ dimensions: 1024, CNN multiplier: 48, dense hidden units: 640, MLP layers: 3). By default, when the input observation $o_t$ is only an image, it is en- and decoded through CNNs. For MiniHack, we have an additional inventory flag that is processed by a separate MLP, as is customary for the Dreamer line of work when dealing with multimodal inputs (Wu et al., 2023). The MLP decoder outputs a Bernoulli distribution from which we sample the decoded inventory flag.

**Reward predictors**  To handle rewards of widely varying magnitudes, DreamerV3 uses twohot codes predicted in symlog space when predicting rewards (Hafner et al., 2023). We use the same setup for all reward prediction heads, i.e., for extrinsic rewards $r_t^i$ for task $i$ or the semantic exploration reward $r_t^{\text{sem}}$. During task-free exploration, the gradients from reward predictions are stopped to not further affect world model training. We do this to keep the world model task-agnostic and to avoid biasing task-free exploration. Similarly, to avoid overfitting to the exploration regime, we also stop the gradients from the semantic reward prediction heads.

**Plan2Explore**  Both our Plan2Explore baseline as well as our ensemble predictors (Eq. 5) are based on the re-implementation on top of DreamerV3. The most notable difference is that in original Plan2Explore the ensemble is trained to predict image encodings (Sekar et al., 2020), whereas the new version is trained to predict stochastic states $z_t$. Recent re-implementations (Hafner, 2021; Hafner et al., 2022; Gumbsch et al., 2024) also used Plan2Explore with ensemble disagreement over $z_t$ as a baseline and verified a strong exploration performance. For our experiments on task-based exploration (Sec. 4.4), we use a weighted sum of extrinsic and intrinsic rewards, i.e. $r_t + \alpha r_t^{\text{dis}}$. We determined the best value $\alpha = 0.5$ through a hyperparameter search, as for SENSEI's hyperparameters.

**Quantile estimation**  We update our estimate of the quantile $Q_k(\hat{r}^{\text{sem}})$ whenever we train the exploration policy. For this, we compute the $k$-th quantile of $\hat{r}_t^{\text{sem}}$ in each training batch ($16 \times 16$). We keep an exponential moving average over these estimates with a smoothing factor of $\alpha = 0.99$.

**Reward weighting**  In practice, we compute exploration rewards (Eq. 7) overall five reward factors:

$$r_t^{\text{expl}} = \chi r_t + \begin{cases} \alpha^{\text{explore}}\hat{r}_t^{\text{sem}} + \beta^{\text{explore}}r_t^{\text{dis}}, & \text{if} \quad \hat{r}_t^{\text{sem}} \geq Q_k(\hat{r}^{\text{sem}}); \\ \alpha^{\text{go}}\hat{r}_t^{\text{sem}} + \beta^{\text{go}}r_t^{\text{dis}}, & \text{otherwise.} \end{cases} \tag{8}$$

i.e. $\chi \in \{0, 1\}$ to scale extrinsic rewards $r_t$, $\alpha$ to scale semantic rewards $\hat{r}_t^{\text{sem}}$ and $\beta$ to scale uncertainty-based rewards $r_t^{\text{dis}}$. We set $\chi = 0$ for task-free exploration and $\chi = 1$ for task-based exploration (Sec. 4.4). When training the value function with DreamerV3, the scale of the reward sources are normalized. To compute this normalization for the exploration policy we use $\alpha^{\text{explore}}$ and $\beta^{\text{explore}}$ of the high percentile region of interestingness ($\geq Q_k$).

### A.2. Semantic Reward Distillation: VLM-MOTIF

For the semantic reward function $R_\psi : \mathcal{O} \to \mathbb{R}$, we use a 2D-convolutional neural network to encode the images. We use 3 convolutional layers, where we progressively increase the number of channels to num_channels_max = 64. The

output then gets downsampled via max pooling before going into a two-layer MLP with hidden dimensions 256 & 512 and outputting the scalar reward value. Additionally, in MiniHack we include inventory information via a separate multi-layer perceptron (MLP) head, consisting of 2 layers with 512 hidden units. The extracted features are concatenated with the image features and get further processed by the output MLP. The training hyperparameters for all $R_\psi$ can be found in Suppl. B.

### A.3. Design Choice: Semantic Reward Predictions

World models typically encode and predict dynamics fully in a self-learned latent state (Ha & Schmidhuber, 2018; Hafner et al., 2023; Hansen et al., 2024). Thus, for a world model to predict $r_t^{\text{sem}}$ at any point in time $t$, we need a mapping from latent states to semantic rewards. We chose to directly predict $\hat{r}_t^{\text{sem}}$ using a reward prediction head of the RSSM. Another option would be to decode the latent state to images and use those as inputs for MOTIF. However, we believe this has several disadvantages: (1) Decoding latent states to images is a computationally costly step that would significantly decrease computational efficiency. (2) We would use an indirect target (the image) instead of the direct target ($r_t^{\text{sem}}$) for training the semantic reward predictions. There would exist no gradient signal to correct somewhat reasonable image predictions that lead to inconsistent reward predictions at a given state. (3) The image predictions of the RSSM can contain artifacts, blurriness or hallucinations. Since MOTIF is only trained on real images from the simulation, we will likely encounter out-of-distribution errors.

## B. Hyperparameters

We provide the hyperparameters used for the world model, exploration policy, VLM-MOTIF annotations & reward model training as well as the environment-specific settings.

| Name | Value | | | |
| --- | --- | --- | --- | --- |
| | Robodesk | KeyRoom | KeyChest | Pokémon Red |
| **World Model** | | | | |
| RSSM size | M | S | S | L |
| Ensemble size $N$ | 8 | 8 | 8 | 8 |
| Train ratio | 512 | 512 | 512 | 512 |
| **Exploration policy** | | | | |
| Quantile | 0.75 - 0.85 -0.75 - 0.80 | 0.90 | 0.90 | 0.6 |
| $\chi$ | 0 - 0 - 0 - 0 | 0 | 0 | 1 |
| $\alpha^{\text{explore}}$ | 0.1 - 0.1 - 0.05 - 0.01 | 0.3 | 0.25 | 0.025 |
| $\beta^{\text{explore}}$ | 1 - 1 - 1 - 1 | 1 | 1 | 0.5 |
| $\alpha^{\text{go}}$ | 1 - 1 - 1 - 1 | 1 | 1 | 0.5 |
| $\beta^{\text{go}}$ | 0 - 0 - 0 - 0 | 0.1 | 0.05 | 0.1 |
| **Annotations for MOTIF** | | | | |
| VLM | GPT-4 turbo (right) & GPT-4 omni (left) | GPT-4 omni | GPT-4 omni | GPT-4 omni |
| Temperature | 0.2 | 0.2 | 0.2 | 0.2 |
| Dataset size | 200K | 100K | 100K | 100K |
| Image res. | $224 \times 224$ | $80 \times 80$ | $80 \times 80$ | $1202 \times 1080$ |
| **MOTIF Training** | | | | |
| Batch size | 32 - 64 - 32 - 32 | 32 | 32 | 32 |
| Learning rate | $10^{-5}$ - $10^{-5}$- $3 \times 10^{-5}$ - $3 \times 10^{-5}$ | $10^{-4}$ | $10^{-4}$ | $10^{-5}$ |
| Weight decay | $10^{-5}$ - 0 - 0 - 0 | $10^{-5}$ | $10^{-4}$ | 0 |
| **Environment** | | | | |
| Action repeat | 2 | 1 | 1 | 1 |
| Episode length | 250 | 600 | 800 | 1k/2k/4k |
| Steps of exploration | 1M | 500K | 500k | 750k |

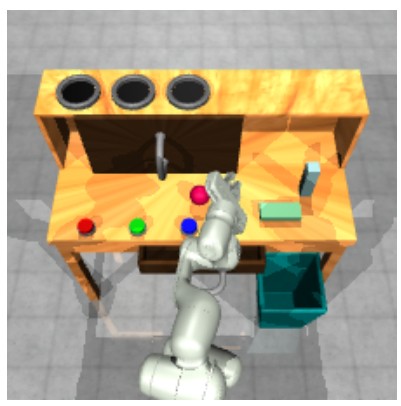

(a) Default observations

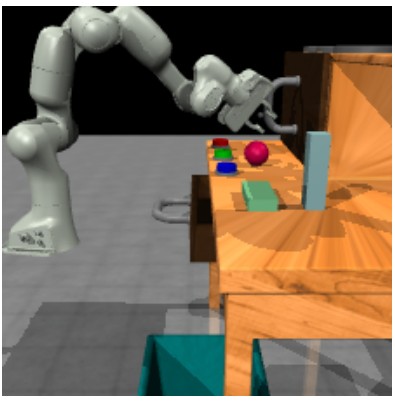

(b) Our observations

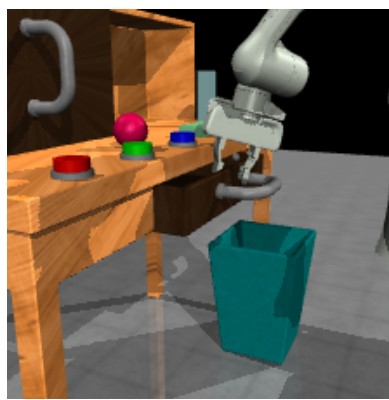

(c) Left camera

*Figure 9.* **Robodesk environment**: We modify the default top-down camera view **(a)** to a side view with less occlusion **(b)**. For annotation with GPT-4 we also provide a left camera observation **(c)**.

For the exploration policy in Robodesk we use different values for the four different variants tested. The values listed here stand for, from left to right: GPT-4 with Plan2Explore (P2X) data using two camera angles for VLM annotations, GPT-4 with P2X data using only the right camera angle, Oracle with P2X data, and Oracle with CEE-US data (corresponding to a more interaction-rich exploration dataset $\mathcal{D}^{\text{init}}$). The VLM-MOTIF hyperparameters are also listed in the same order.

**Image resolution**  For the world model we use $64 \times 64$ pixel images for all environments. However, for the GPT annotations we use higher resolution images, as shown in the table. Inside the environment `step` function, the rendering is performed at these higher resolutions, and this image is input to the semantic reward function $R_\psi$. The image is then scaled down to $64 \times 64$ as part of the observation that the RSSM is trained on.

**Baselines**  We run DreamerV3 with the same world model setup as SENSEI and Plan2Explore. We use an open source PPO (Schulman et al., 2017) implementation of Hafner (2024)[4] optimized to work well across multiple environments with a fixed set of hyperparameters (details in Hafner et al., 2023, supplementary material). We build our RND (Burda et al., 2019) implementation on top of PPO. For the predictor and target network we use a ResNet with 3 convolutional layers followed by 5 dense layers. We only use the intrinsic reward to train a PPO agent. Intrinsic rewards are normalized as outlined in Burda et al. (2019). While Burda et al. (2019) also normalize input observations through a running statistics, we found that using LayerNorm at the input layer leads to slightly more interactions in Robodesk.

## C. Environment Details

### C.1. Robodesk

Robodesk (Kannan et al., 2021) is a multi-task RL benchmark in which a robot can interact with various objects on a desk. We use an episode length of 250 time steps.

**Observations**  Robodesk uses only an image observation, depicting the current scene, which we scale down ($64 \times 64$ pixels). However, we found that the default top-down view often had occlusions and was hard to interpret from a single image (Fig. 9a). Thus, we used a different camera angle showing the robot from one side (Fig. 9c). With this view objects and the drawer were rarely occluded; however, lights that turn on from button presses were not as visible anymore.

**Actions**  The continuous 5-dimensional actions control the movement of the end effector. We use an action repeat of 2 to speed up the simulation. Thus, 1M steps of exploration correspond to 2M actions in the environment.

**Interaction metrics**  We track how often the robot interacted with different objects to quantify the behavior during exploration. Specifically, we track the velocity of joints and object positions. For buttons, sliding cabinet, or drawer, we

---

[4]https://github.com/danijar/embodied, version v1.2

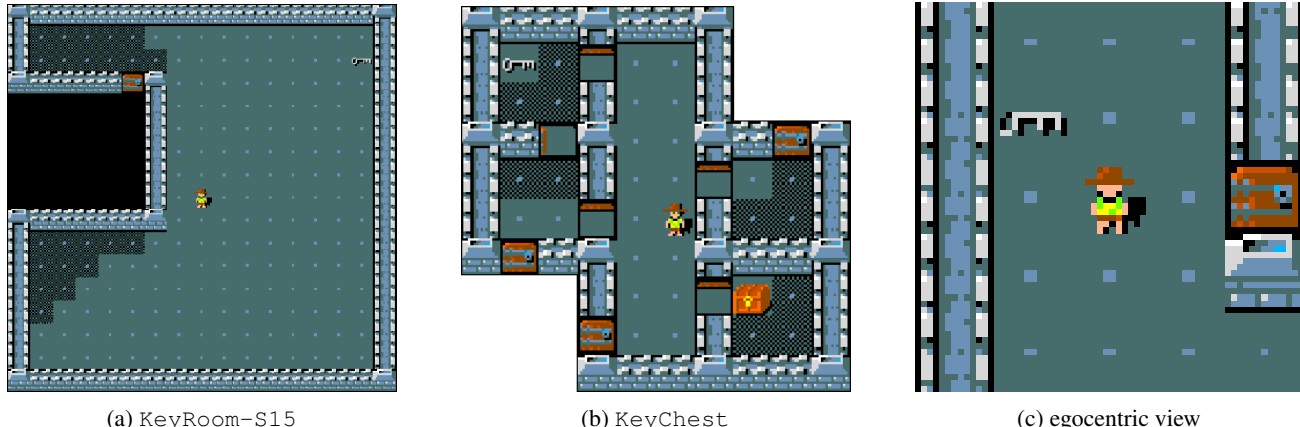

(a) `KeyRoom-S15`          (b) `KeyChest`          (c) egocentric view

*Figure 10.* **MiniHack** : We consider two tasks `KeyRoom-S15` **(a)** and `KeyChest` **(b)**. The agent receives an egocentric view of the environment as its observation **(c)**.

check if the joint position changes more than a fixed value (0.02). For all other objects, we check if any of their $x$-$y$-$z$ velocities exceed a threshold (0.02).

**Tasks**   We use the sparse reward versions of all the tasks available in the environment. For some tasks, we add easier versions. All tasks describe interactions with one or multiple objects:

- **Buttons**: Pushing the red (`push_red`), blue (`push_blue`), or green (`push_green`) button.
- **Sliding cabinet**: Opening the sliding cabinet fully (`open_slide`).
- **Drawer**: Opening the drawer fully (`open_drawer`), opening the drawer half-way (`open_drawer_medium`), or opening it slightly (`open_drawer_light`). We introduced the latter tasks.
- **Upright Block**:  Lifting  the  upright  block  (`lift_upright_block`),  pushing  it  off  the  table (`upright_block_off_table`) or putting it into the shelf (`upright_block_in shelf`).
- **Flat Block**: Lifting the flat block (`lift_flat_block`), pushing it off the table (`flat_block_off_table`), into the bin (`flat_block_in_bin`), or into the shelf (`flat_block_in_shelf`).
- **Both blocks**: Stacking both blocks (`stack`).
- **Ball**:  Lifting  the  ball  (`lift_ball`),  dropping  it  into  the  bin  (`ball_in_bin`)  or  putting  it  into  the  shelf (`ball_in_shelf`).

### C.2. MiniHack

**Observations**   In MiniHack multiple observation and action spaces are possible. We use egocentric, pixel-based observations centered on the agent ($\pm2$ grids, example in Fig. 10c). In addition to that, we provide the agent's inventory. By default, in MiniHack the inventory is given as an array of strings (UTF8 encoded), and different player characters have different starting equipment based on the character classes of NetHack. We simplify this by providing only a binary flag that indicates if the agent has picked up a new item. This is sufficient for the problems we consider, in which maximally one new item can be collected and starting equipment cannot be used.

**Environments**   Here we detail the environments we tackle:

In the benchmark `KeyRoom-S15` problem (Fig. 10a), the agent needs to fetch a key in a large room ($15 \times 15$ grids) to enter a smaller room and find a staircase to exit the dungeon. We use the default action space but enable autopickup and therefore remove the `PICKUP` action. We use an episode length of 600 time steps, which is 1.5 times longer than the default episode length.

`KeyChest` is a novel environment designed by us, based on `KeyCorridorS4R3` from MiniGrid (Chevalier-Boisvert et al., 2024) (see Fig. 10b). The agent starts in a corridor randomly connected to different rooms. A key is hidden in one room and a chest in another room. The goal is to open the chest with the key in the inventory. Object positions are

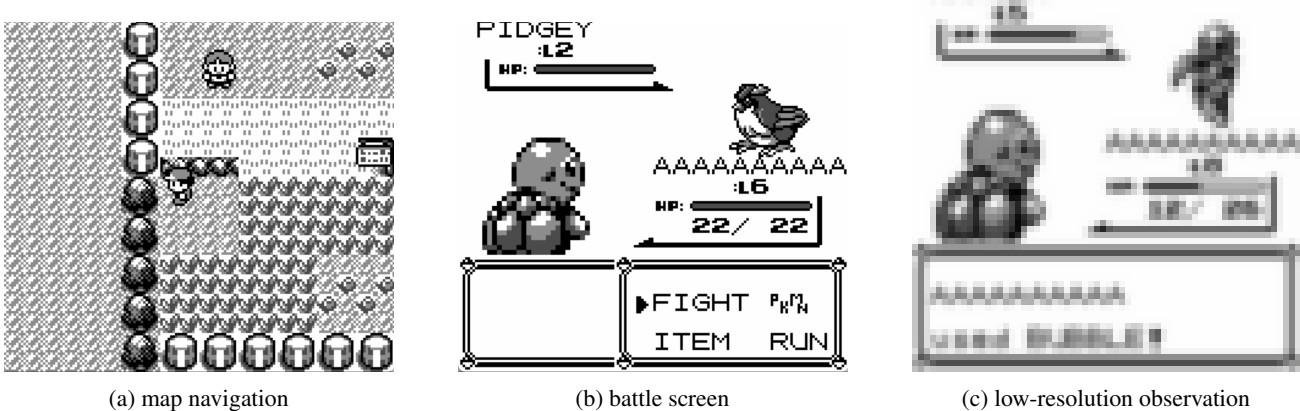



(a) map navigation        (b) battle screen        (c) low-resolution observation



*Figure 11.* **Pokémon Red** poses a strong exploration challenge as it requires an agent to learn (1) to navigate a complex overworld map **(a)** and (2) to battle and catch Pokémon **(b)**. We only use the down-scaled game screen image as observations **(c)**.

randomized. The action space for this task contains 5 discrete actions for moving the agent in 4 cardinal directions (UP, RIGHT, DOWN, LEFT) and an OPEN-action to open a chest when standing next to it with a key in the inventory. Episodes terminate when the chest is opened. We enable auto-pickup, so no additional action is needed to pick up the key when stepping on it. We use an episode length of 800 time steps.

**Rewards** All environments use a sparse reward of $r_t = 1$, which the agent only receives upon accomplishing the task. A small punishment ($r_t = -0.01$) is given, when the agent performs an action that does not alter the screen.

**Image remapping** Empirically, we found that GPT-4 may encounter problems if we provide the image observations as is. For example, when using the default character in the KeyRoom-S15 environment (Rogue), GPT-4 sometimes throws content violation errors. We suspect that this is due to the character wearing a helmet with horns, which could be mistaken for demonic or satanic imagery. Thus, we pre-processed the images before returning them from the environment. We render all characters as the Tourists, a friendly looking character with a Hawaiian shirt and straw hat. Furthermore, GPT-4 sometimes mistakes entrance staircases for exit staircases. Since the entrance staircases serve no particular purpose and are not different from the regular floor, we remap all entrance staircases to floors.

### C.3. Pokémon Red

We evaluate SENSEI's ability for semantic exploration in the classic Game Boy game, Pokémon Red. Pokémon Red presents an extremely challenging exploration problem due to: (1) its vast and interconnected world, composed of towns, routes, forests, and other areas that must be navigated, and (2) its complex battle system, which requires semantic knowledge to understand type interactions (e.g., Water attacks are strong against Fire-type Pokémon). Thus, an agent needs strong exploration capabilities to progress in the game's primary objective, i.e., becoming the Pokémon Champion by defeating Gym Leaders and collecting their badges. We base our implementation on PokeGym v0.1.1 (Suarez, 2024), which is based on Whidden (2023), with minor modifications as detailed below.

**Observations** Unlike previous RL agents applied to Pokémon (Whidden, 2023; Pleines et al., 2025), we provide only the raw game screen image as input, without access to the internal game state or additional memory. When input into the world model, we downscale the game screen (to 64x64 pixels) in order to save compute (see Fig. 11c). For VLM annotations, we use the original size (1202x1080 pixels), such that all text is clearly readable (see Fig. 11b).

**Actions** The agent controls the game using a 6-dimensional action space corresponding to Game Boy button presses (Left, Right, Up, Down, A, B). Since the game only advances upon button presses (except during attack animations) we only apply an action every 1.5 seconds real time game play (frame skip of 96) to manage episode length.

**Rewards**    The agent receives a weighted sum of rewards $r_t = \sum \phi^i r_t^i$ for different in-game events $i$. Rewarded events include leveling up, catching Pokémon, encountering strong opponents, healing Pokémon, visiting new map tiles, earning badges, and a penalty for blacking out after losing a battle. We leave the default values $\phi^i$ from Suarez (2024) except we increase $\phi^{\text{tiles}} = 0.1$ for reaching new map tiles (previously set to $0.01$), as we found this improves exploration.

**Episode Length**    To scaffold exploration, we gradually increase the maximum episode length over environment steps: 1k length for 0–25k steps, 2k length for 25–50k, 4k length for 50–75k, and 8k length when continuing exploration beyond 75k steps.

**Starting point**    As in Whidden (2023), we skip the initial "tutorial" phase of the game, and start only when the player can freely move and catch Pokémon. In-game, this corresponds to the point after delivering Oak's Parcel and receiving the Pokédex and Poké Balls from Professor Oak. We use the same checkpoint as Whidden (2023), starting with a level 6 Squirtle named AAAAAAAAAA.

**Map segments**    The accessible game world in Pokémon Red prior to defeating the first Gym is already expansive, comprising a variety of interconnected areas. Many of these are distinct maps that load separately when the player enters or exits a building (see Fig. 12). To evaluate spatial exploration, we segment this world into discrete map segments, as illustrated in Fig. 8a. Each route and town is treated as a separate segment, as are buildings or enclosed areas like forests that load a dedicated sub-map. This amounts to 25 map segments that can be accessed before beating the first Pokémon Gym. Reaching the first Pokémon Gym requires navigating through 10 such segments, which we enumerate in order of appearance (cf. Fig. 8a):

| Number | Name | Type |
|--------|------|------|
| 0 | Oak's Lab | building |
| 1 | Pallet Town | town |
| 2 | Route 1 | route |
| 3 | Viridian City | town |
| 4 | Route 2 | route |
| 5 | Viridian Forest South Gate | building |
| 6 | Viridian Forest | forest |
| 7 | Viridian Forest North Gate | building |
| 8 | Pewter City | town |
| 9 | Pewter Gym | building |

This segmentation allows us to quantify exploration progress by measuring the highest-indexed map segment reached during an episode.

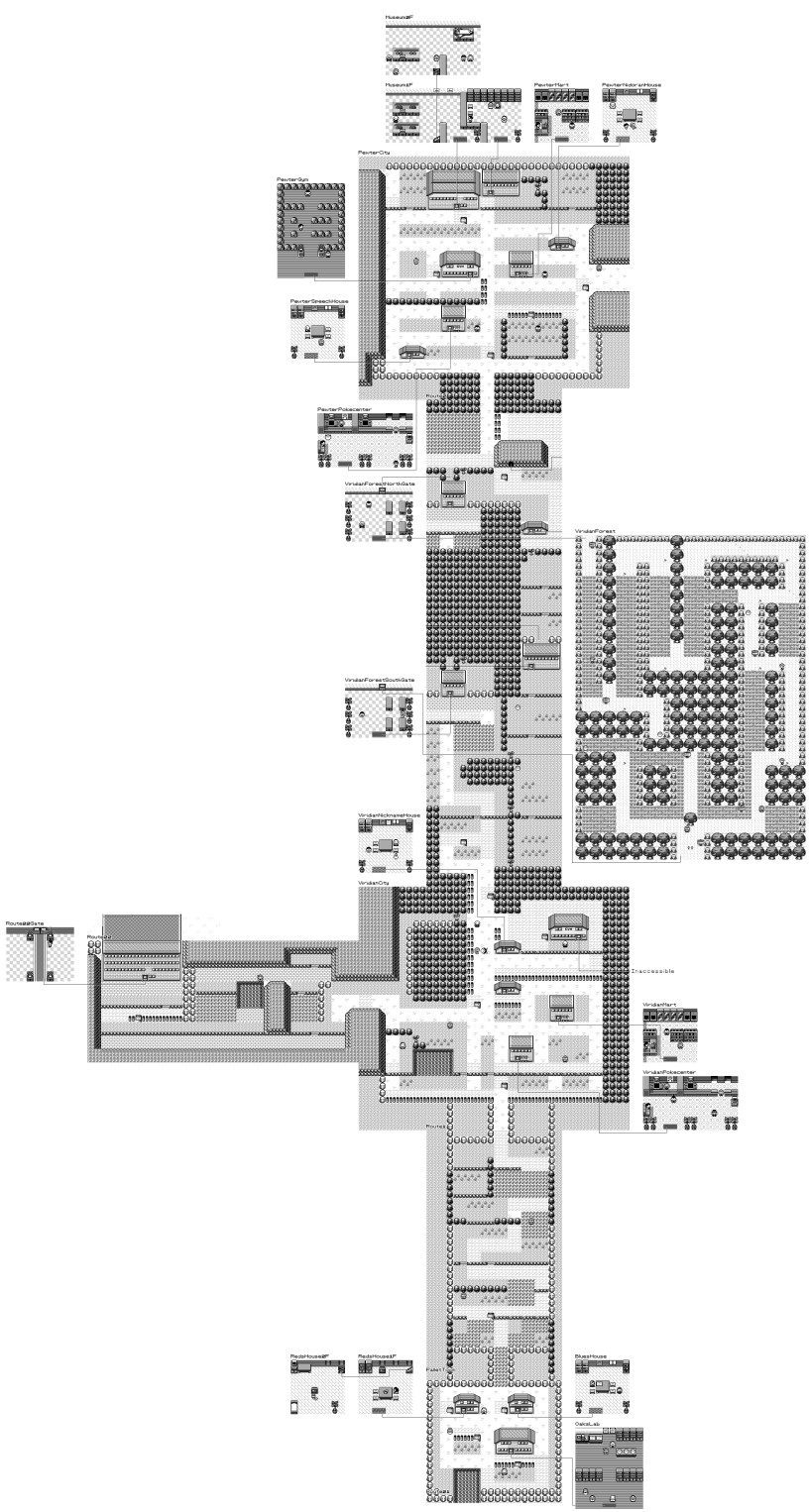

*Figure 12.* **Full map of Pokémon Red** accessible prior to defeating the first Gym. If entering or exiting a building brings the agent to a submap, this is indicated by a line. The map is modified from https://blog.vjeux.com/2023/project/pokemon-red-blue-map.html

# D. VLM prompting

We prompt the VLM with somewhat general descriptions of the environments that we consider. Here we provide the full prompts for all environments.

## D.1. Robodesk

In Robodesk, for each query, we provide two observation images (resolution $224 \times 224$) with the following prompt:

```
Here are two images in a simulated environment with a robot in front of a
desk.  Your task is to pick between these images based on how interesting they
are.  Which image is more interesting in terms of the showcased behavior?  For
context following points would constitute interestingness:  (1) The robot
is currently holding an object in its gripper.  (2) The robot is pushing
an object around or pushing a button or opening the drawer or interacting
with entities on the desk.  (3) Objects on the desk are in an interesting
configuration: e.g.  a stack.  Being far away from the desk with the robot
arm retracted or just stretching your arm without interactions, is a sign the
image is not interesting.  Answer in maximum one word:  0 for image 1, 1 for
image 2, 2 for both images and 3 if you have no clue.
```

Due to occlusions, we annotate the same pair from the initial dataset $\mathcal{D}^{\text{init}}$ with the same prompt using images from two camera angles: right (Fig. 9c) and left (Fig. 9c). A pair is deemed valid only if the GPT-4 response is the same across both camera angles, otherwise the pair and the annotation are removed from the dataset. The VLM-MOTIF training as well as the world model training are still executed using only the right camera image, such that during free play with SENSEI we only rely on the right camera images as input.

In Robodesk, we use GPT-4 Turbo (`gpt-4-turbo-2024-04-09`) for the right camera annotations and GPT-4 omni (`gpt-4o-2024-05-13`) for the left camera annotations. To cap the dataset size of pairs at 200K and still make sure that we got contrastive pairs of interesting vs. uninteresting behaviors, we pre-filtered the pairs using our Oracle of interestingness for the Plan2Explore dataset. We believe that in future work this step can be omitted by simply generating more random pairs and expanding the dataset size. After the two-angle image annotation, we use the subset of the dataset where both left and right camera annotations agree. This corresponds to 139409 pairs from the original 200K dataset, or 69.7% of it.

## D.2. Robodesk: Zero Pre-Knowledge General Prompt

To demonstrate the generality of SENSEI, we also test a version with a general prompt (SENSEI GENERAL), using a multi-turn strategy for getting the annotations for a pair of images. This version of the prompt does not contain any external environment specific knowledge. First, we show a picture from the robotic environment (shown in Fig. 9c) and ask the VLM (`gpt-4o-2024-05-13`) to describe what it sees in the scene using the following prompt:

```
I will show you an image from a simulated environment.  Describe the
environment to me and what you deem the agent and then explain what types
of behavior / observations would be useful and interesting for the agent to
explore in this setting?
```

We obtain the following response from GPT:

```
The image depicts a simulated environment featuring a robotic arm interacting
with a table that has various objects on it.  The objects include buttons
of different colors, a red ball, a green cube, and a blue rectangular block.
The setting appears to be designed for testing robotic manipulation and
interaction with objects.
Agent:  The robotic arm is the agent in this environment.  It is likely
programmed to interact with the objects on the table, possibly for tasks
involving sorting, stacking, or pressing buttons.
Useful and Interesting Behaviors/Observations:        1.  Object Manipulation:
 - Grasping and moving objects like the ball, cube, and block.  - Testing
different grips and force applications to handle objects of various shapes
and weights.  2.  Button Interaction:  - Pressing buttons to observe effects,
such as changes in the environment or triggering specific actions.  -
Learning which buttons correspond to which actions or outcomes.  3.  Spatial
Awareness:  - Navigating the arm around the table without knocking over
objects.  - Understanding the spatial relationship between objects for
tasks like stacking.  4.  Task Learning:  - Completing specific tasks
like sorting objects by color or shape.  - Learning sequences of actions
to achieve complex goals.  5.  Feedback and Adaptation:  - Using sensory
feedback to adjust actions in real-time.  - Adapting to changes in the
environment, such as objects being moved or removed.  6.  Exploration
and Experimentation:  - Experimenting with different strategies for
interacting with objects.  - Exploring the effects of different actions on
the environment.  These behaviors would help the agent develop skills in
manipulation, decision-making, and adaptive learning, which are crucial for
more complex real-world applications.
```

Next, we show the pair of images that we would like GPT-4 to annotate with the following prompt:

```
Now I will show you two images.  Based on your answer, choose which one of
these images is more interesting in terms of the showcased behavior?  Answer
in maximum one word:  0 for image 1, 1 for image 2, 2 for both images and 3 if
you have no clue.
```

We obtain the final annotation for the pair as the answer to this last prompt.

In this experiment, the environment description is generated using the first prompt only once. For subsequent annotations, the initial question-answer pair is included in GPT-4's context window as history. In the second turn, the new image pair is presented using the final prompt. This approach is more cost-efficient, as in-context tokens are less expensive than generated ones, and time-efficient, as generating the full environment description sequentially requires more processing time. With this multi-turn strategy, annotating each image pair costs approximately $0.004, compared to $0.002 per annotation with the single-turn prompt.

### D.3. MiniHack

For MiniHack we provide two observation images (resolution $80 \times 80$) and the inventory. For the inventory, we only consider items that were picked up and not items in the agent's starting equipment[5]. We provide the inventory as text descriptions. The different options are shown in purple.

---

[5]The starting equipment is taken from the NetHack game and irrelevant and inaccesible in our tasks.

```
Your task is to help play the video game MiniHack.  MiniHack is a roguelike
game where an agent needs to navigate through rooms and escape a dungeon.  For
succeeding in the game, finding items, collecting items and exploring new
rooms is crucial.  Images are egocentric around the agent, who is standing on
a dotted blue floor.  Your task is to pick between two game states, composed
of images and an inventory descriptions, based on how interesting and useful
they are.
Is there any difference between the first and second game state in terms of
how interesting it is?  The images depict the current view.  {The first agent
has a key named The Master Key of Thievery in their inventory., The second
agent has a key named The Master Key of Thievery in their inventory., Both
agents have a key named The Master Key of Thievery in their inventory., Both
agents have no items in their inventory.},
Think it through and then answer in maximum one word:  0 if the first state is
more interesting, 1 if the second state is more interesting, 2 if both states
are interesting and 3 if nothing is interesting or you are very unsure.
```

For MiniHack we use GPT-4 omni (`gpt-4o-2024-05-13`).

### D.4. Pokémon Red

For Pokémon Red we set the goal to reach the first Pokémon Gym (the first boss battle of the game): We assume that GPT-4 (`gpt-4o-2024-05-13`) was extensively trained on game play data and various walkthroughs of Pokémon Red and contains sufficient knowledge of the game. Thus, we again use a multi-turn strategy for image annotations, as with SENSEI GENERAL (see Suppl. D.2), where we first ask the VLM for a game play description, which we then use as context for further annotations. We first asked the VLM:

```
Your task is to help me play the Game Boy game Pokémon Red.  I just obtained
my starter Pokémon, Squirtle.  My goal is to find and defeat the Gym Leader,
Brock.  What do I need to do, and which areas do I need to traverse to achieve
this goal?  Keep it very short.
```

We generated five different responses from GPT-4. We sample from them uniformly as context for image-based comparisons. We provide two observation images with the following prompt:

```
Here are two screenshots from the game.  Which image depicts a game state that
is closer to my goal?  Answer in maximum one word:  0 if the first state is
better, 1 if the second state is better.
```

### D.5. Oracle for Interestingness

In Robodesk, we also use an Oracle of interestingness to annotate the pairs as an ablation (see Suppl. E.2). Our goal here is to showcase an upper-bound of performance on SENSEI without the noisiness of VLMs. For the Oracle, we deem a state interesting if: (1) any one of the entities are in motion (here only for the ball we make an exception that the ball should be in motion with the end effector close to it as the ball in the environment is unimpeded by friction), (2) if the drawer is opened, (3) if the drawer/sliding cabinet is not yet in motion, but the end effector is very close to their handles, (4) if the upright and flat blocks are not yet in motion but the end effector is very close to them (almost touching), (5) if the stacking task is solved. With these statements, we essentially cover the range of tasks defined in the Robodesk environment, as they are shown in Fig. 16.

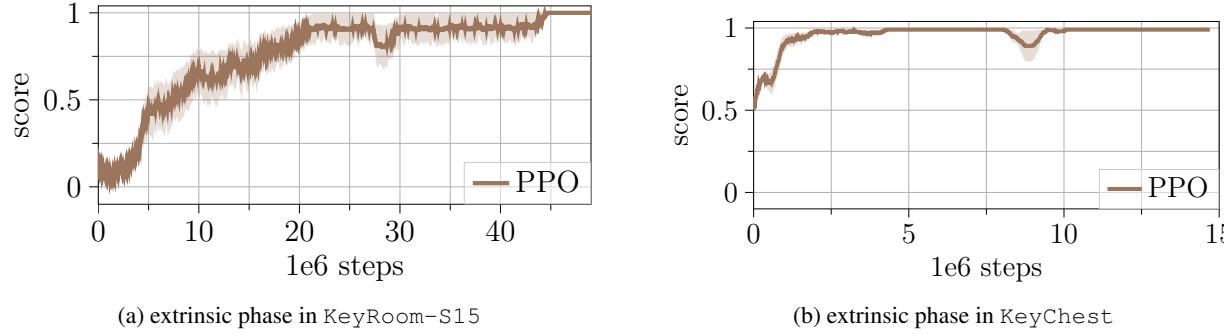

(a) extrinsic phase in `KeyRoom-S15`

(b) extrinsic phase in `KeyChest`

*Figure 13.* **PPO performance in MiniHack**: We plot the mean episode score obtained by PPO during evaluation for the MiniHack tasks `KeyRoom-S15` **(a)** and `KeyChest` **(b)**. Shaded areas depict the standard error (10 seeds). We apply smoothing over the score trajectories with window size 30.

## E. Extended Results

### E.1. MiniHack: Extended Results

Figure 13 shows the full trajectory of evalutation scores for Proximal Policy Optimization (PPO, Schulman et al. 2017) in MiniHack when trained until convergence. While PPO manages to learn to solve all tasks, it can be much less sample efficient than the model-based agents we evaluated (see Fig. 7), especially in `KeyRoom-S15`. Here SENSEI outperforms PPO in terms of sample efficiency in one to two orders of magnitude.

### E.2. Robodesk: SENSEI Ablations

In Robodesk, we compare different versions of SENSEI in order to analyze the effect of the VLM and the initial exploration data on SENSEI performance (Fig. 14). First, we showcase SENSEI results when annotating the initial exploration dataset from Plan2Explore with only the right camera images. In this case, we use the whole 200K pairs in the dataset, without any pruning. In another ablation, we replace the VLM (GPT-4) with a hand-crafted Oracle (see Suppl. D.5 for how the oracle is computed) for annotating the pairs. After the oracle annotations, we distill these preferences into VLM-MOTIF for SENSEI, following the same procedure as before. Furthermore, we compare two initial datasets $\mathcal{D}^{\mathrm{init}}$ of self-supervised exploration collected either by CEE-US (Sancaktar et al., 2022) or by Plan2Explore for the oracle SENSEI versions. CEE-US uses vector-based position of entities for information-gain-based exploration, in comparison to Plan2Explore, which works on the pixel-level. Due to the privileged inputs, $\mathcal{D}^{\mathrm{init}}_{\mathrm{CEE-US}}$ contains more complex interactions. We compare 1M steps of exploration with the four versions of SENSEI and Plan2Explore.

On average, all versions of SENSEI interact more with the objects than Plan2Explore and our semantic exploration reward seems to lead to more object interactions than pure epistemic uncertainty-based exploration. SENSEI with Oracle for both the Plan2Explore and especially the CEE-US initial datasets show the most object interactions. We believe this further

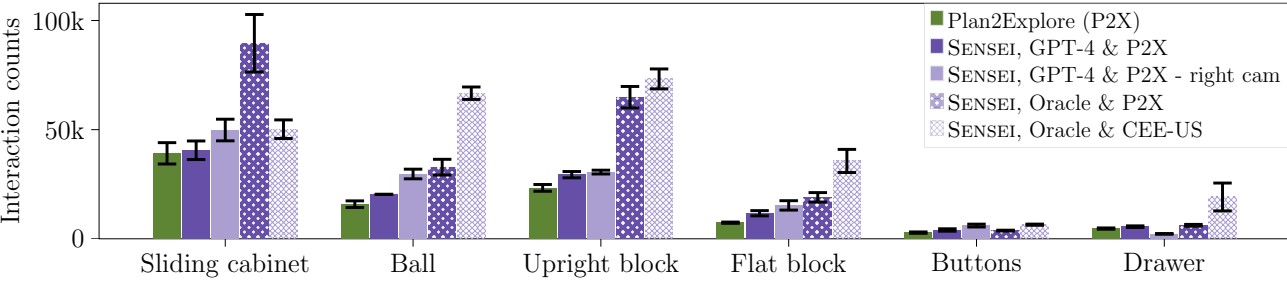

*Figure 14.* **Interactions in Robodesk**: We plot the mean over the number of interactions with objects in the environment during exploration for different versions of SENSEI (Oracle vs. VLM, CEE-US (Sancaktar et al., 2022) vs. Plan2Explore to create the data to label $\mathcal{D}^{\mathrm{init}}$) and Plan2Explore. We also ablate SENSEI using only the right camera angle for VLM annotations on the Plan2Explore dataset. Error bars show the standard deviation (3 seeds).

showcases that the VLM provides a much noisier signal of interestingness, making it harder to optimize for.

The initial exploration dataset $\mathcal{D}^{\text{init}}$ influences with which objects SENSEI interacts. Qualitatively, we observe Plan2Explore performing mostly arm stretches. Interestingly, this can still lead to solving tasks during exploration. For example, stretching the arm against the sliding cabinet can close it, and stretching the arm toward the upright block can push it off the table. As a result, SENSEI with Plan2Explore Oracle focuses mainly on the sliding cabinet and the upright block, reinforcing the existing trends in the initial dataset from which VLM-MOTIF is distilled.

For CEE-US data, Oracle SENSEI interacts more with the other objects, such as the ball and the flat block, as well as the drawer. The difference between the Oracle annotator SENSEI versions with CEE-US vs. Plan2Explore data showcases that there is still a lot to be gained from a richer initial dataset for SENSEI, which could be obtained via multiple rounds of SENSEI exploration.

If a VLM annotates images instead of the Oracle, SENSEI shows similar behavioral trends, but overall less object interactions, such that neither of the GPT-4 annotations on the Plan2Explore data completely match the performance of the oracle annotator.

Finally, when we compare the performance for SENSEI using GPT-4 annotations with two-angle camera images vs. only the right camera angle image, we see that the two-angle version performs better in terms of drawer interactions. This is expected since the drawer is more clearly visible in the left camera view. However, as the ball and blocks are mainly initialized on the right side of the table, the pure right camera angle SENSEI generates more interactions with these objects during exploration. Another factor here is that for the right camera angle we retain all 200K pairs for VLM-MOTIF distillation, whereas we only keep ca. 70% of the pairs in the case of SENSEI using both cameras for annotation.

### E.3. Robodesk: Rewards

In addition to interaction metrics, we count the number of times task rewards are collected during exploration. We observe that for the majority of tasks SENSEI solves more tasks in the environment during play than Plan2Explore. Note that for the `open_slide` task you need to open the slide fully in one direction, which is achieved in abundance in Plan2Explore runs by simply stretching the arm. The full interaction metrics of exploring how the slide moves left-right is not necessarily reflected in the task rewards, as can be seen in comparison to Fig. 6. Similar arguments also apply for opening the drawer fully vs. opening and closing the drawer more dynamically. Additionally as the bin is not really visible in our camera angle, solving `in_bin` tasks are more due to the objects that go off the table landing by chance in the bin for all methods, such that higher statistics for `off_table` rewards also lead to higher `in_bin` rewards.

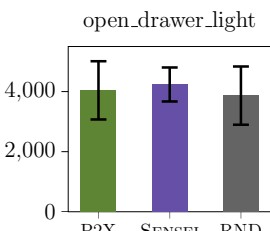

*Figure 15.* **Collected rewards** for `open_drawer_light` during exploration (3 seeds).

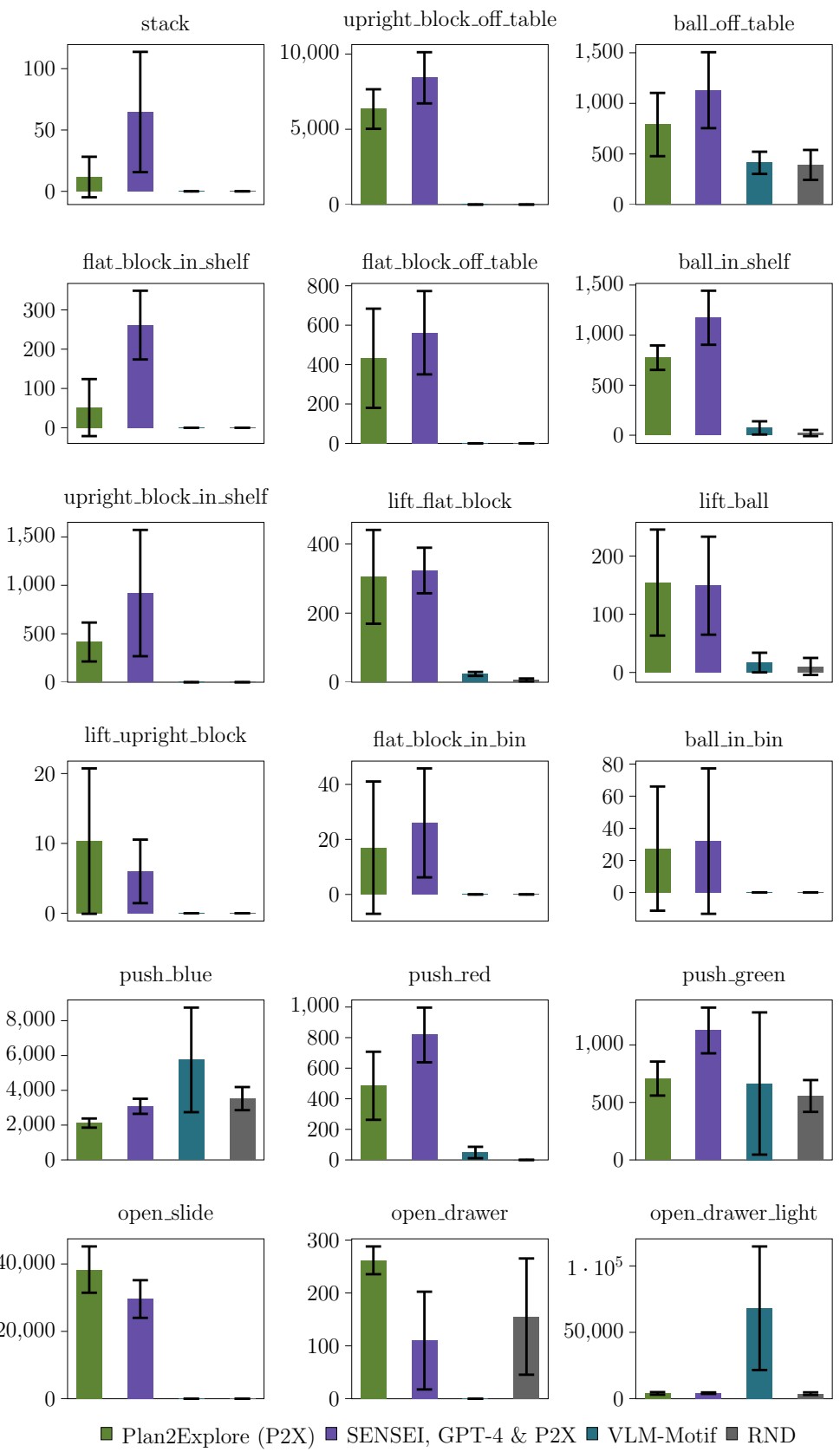

*Figure 16.* **Robodesk environment rewards**: We plot the mean number of sparse rewards (successful task completions) discovered during 1M steps of task-free exploration for all tasks for Plan2Explore, SENSEI, pure VLM-MOTIF, and the RND baseline.

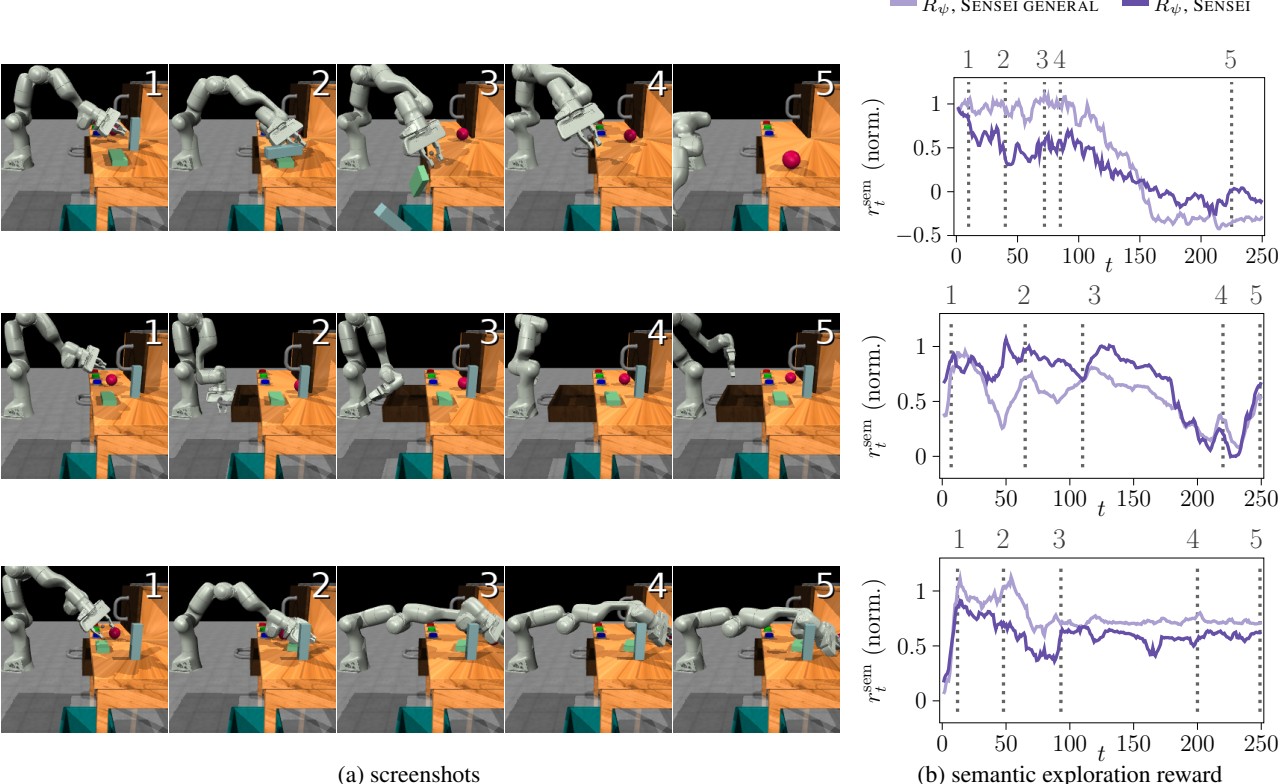

(a) screenshots        (b) semantic exploration reward

*Figure 17.* **Semantic exploration rewards for example trajectories with VLM-MOTIF using general vs. specialized prompts**: For three example Robodesk episodes, we showcase VLM-MOTIF semantic rewards distilled from GPT-4 annotations using a prompt specialized to the environment vs. a general prompt using multi-turn annotations. The reward trajectories for both the general and specialized prompts peak at the "interesting" moments of exploration, such as opening a drawer or pushing the blocks. With zero external knowledge injection, the general prompt version of VLM-MOTIF is highly correlated with its specialized prompt counterpart.

### E.4. Robodesk: VLM-MOTIF with General Prompt

In this section, we investigate the distilled reward function when using a general prompting strategy (SENSEI GENERAL, see Suppl. D.2). As shown in Fig. 17, the semantic reward $r_t^{\text{sem}}$ for the general prompt seems to show a high positive correlation or qualitatively matches with the VLM-MOTIF distilled using the specialized prompt in Robodesk (see Suppl. D.1). Thus, we manage to distill a reward function that peaks at interesting moments of exploration without injecting any environment specific knowledge into the prompt.

### E.5. Robodesk: Baselines

We present two other baselines in Robodesk: RND trained with PPO and pure VLM-MOTIF, and analyze the interaction metrics in Fig. 18. On average, SENSEI interacts more with most available objects than the baselines. RND mostly moves the arm around in the center of the screen, occasionally hitting objects or mostly buttons. It is important to note that the robot arm in Robodesk is mostly initialized close to the buttons. Pure VLM-MOTIF is an ablation of SENSEI without any information gain objective. Here, we see the importance of the information gain reward to ensure diverse exploration. Unlike SENSEI, we see that VLM-MOTIF interacts with specific entities: mostly the buttons, the drawer and the flat block. The lack of interaction with the cabinet, the upright block and the ball are expected as these entities are spatially further away from the robot initialization pose. Once high semantic rewards are found in the vicinity by interacting with the drawer and buttons, there is no incentive for pure VLM-MOTIF to explore further. On the other hand SENSEI aims to discover interesting and yet novel behaviors, ensuring better coverage across the different useful behaviors in the environment.

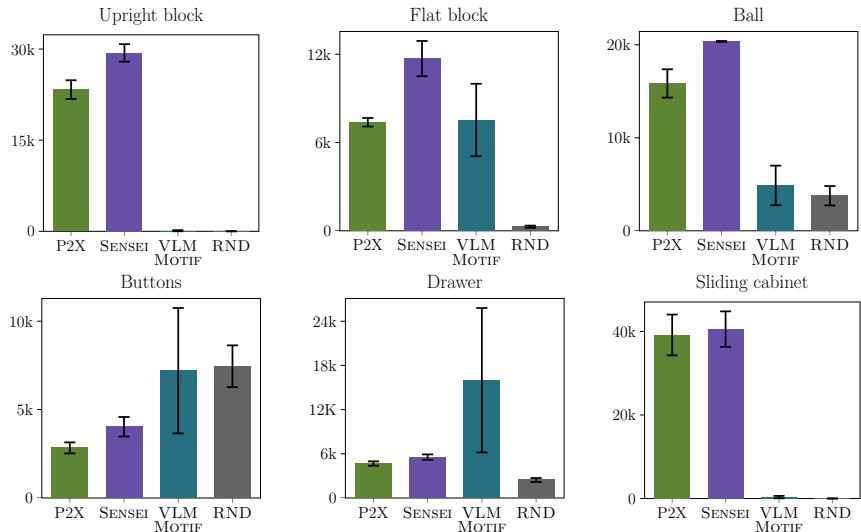

*Figure 18.* **Interactions in Robodesk**: We plot the mean over the number of interactions with any object during 1M steps of exploration for SENSEI, Plan2Explore (P2X), pure VLM-MOTIF and Random Network Distillation (RND) trained with a PPO policy, as a model-free exploration baseline. Error bars show the standard deviation (3 seeds).

### E.6. Robodesk: SENSEI without Dynamic Scaling and Analyzing Hyperparameter Sensitivity

In this section, we ablate the dynamic scaling of the semantic reward $r_t^{\text{sem}}$ and the information gain reward $r_t^{\text{dis}}$ terms in SENSEI. In SENSEI, we adjust the weight of these two terms based on whether $r_t^{\text{sem}}$ has reached the high percentile region of interestingness ($r_t^{\text{sem}} \geq Q_k$), as per equation Eq. 8. In this ablation, we instead use a linear combination with fixed weights $\alpha$ and $\beta$, such that the exploration reward is given by:

$$r_t^{\text{expl}} = \alpha r_t^{\text{sem}} + \beta r_t^{\text{dis}}. \tag{9}$$

We present the results in Fig. 19 for 6 different sets of fixed weights. First of all, we observe that none of the fixed scale settings outperform SENSEI nor do they consistently perform as well as SENSEI. Second of all, we see that the exploration behavior is very sensitive to the choice of the weights $\alpha$ and $\beta$. For larger $\alpha$ values, the behavior collapses to mostly interacting with the drawer, buttons and the flat block, with larger fluctuations. This mode is very similar to the case of pure VLM-MOTIF presented in Fig. 18.

Next, we test the hyperparameter sensitivity of SENSEI with dynamic scaling of the reward weights. We see in Fig. 20, that across all 4 hyperparameter configurations, SENSEI is better or at least on par with Plan2Explore, and we don't observe any behavior collapse as in the fixed scale setting. We argue that although the dynamic scaling introduces additional hyperparameters, the overall behavior is much more robust and less dependent on hyperparameter tuning.

*Table 1.* Hyperparameter configurations for SENSEI presented in the main experiments and the 3 other configurations that are shown in Fig. 20.

|  | SENSEI | SENSEI HP1 | SENSEI HP2 | SENSEI HP3 |
| --- | --- | --- | --- | --- |
| Quantile | 0.75 | 0.80 | 0.85 | 0.75 |
| $\alpha^{\text{explore}}$ | 0.1 | 0.01 | 0.1 | 0.05 |
| $\beta^{\text{explore}}$ | 1 | 1 | 1 | 1 |
| $\alpha^{\text{go}}$ | 1 | 1 | 1 | 1 |
| $\beta^{\text{go}}$ | 0 | 0 | 0 | 0 |

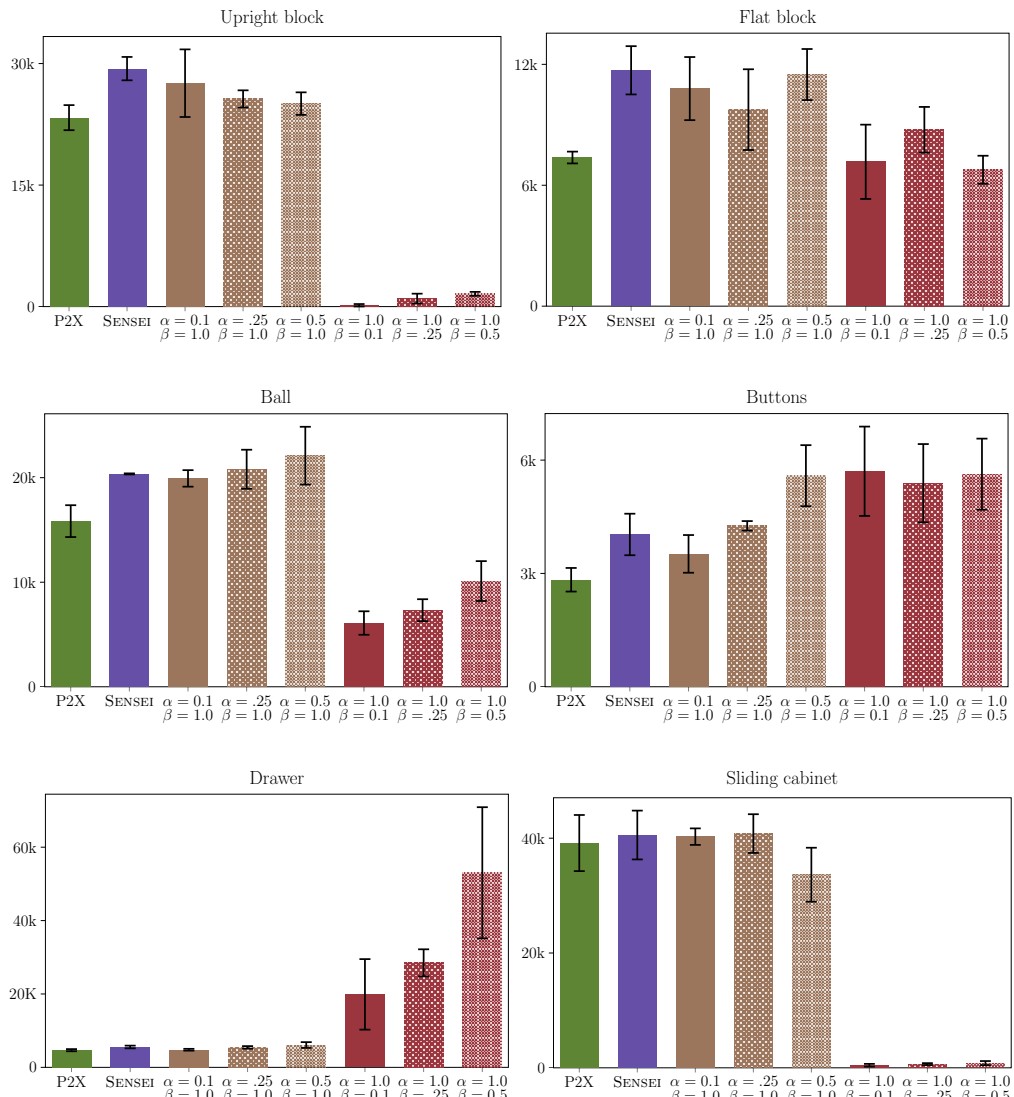

*Figure 19.* **Comparing interactions in Robodesk between SENSEI and fixed scaling of rewards**: We plot the mean over the number of interactions with any object during 1M steps of exploration for SENSEI and Plan2Explore (P2X) and an ablation of SENSEI, where we do not dynamically adjust the weight of the reward terms based on the current semantic reward. For this ablation, reward is computed as $r_t^{\text{expl}} = \alpha r_t^{\text{sem}} + \beta r_t^{\text{dis}}$ with fixed weights $\alpha$ and $\beta$. Error bars show the standard deviation (3 seeds).

### E.7. Robodesk: Downstream task learning

We investigate whether the world model learned by SENSEI is versatile enough to efficiently support downstream task learning in the multi-task environment of Robodesk. We evaluate this on three representative tasks: opening the drawer (`Drawer-Medium`), pushing the upright block off the table (`Upright-Block-Off-Table`), and lifting the ball (`Lift-Ball`).

As in our MiniHack experiments (see Sec. 4.3), we initialize a DreamerV3 agent with a pre-trained world model obtained from 1M steps of SENSEI-driven exploration. We compare this setup to exploration with Plan2Explore, running each agent for 1.2M steps or until SENSEI converges on the task.

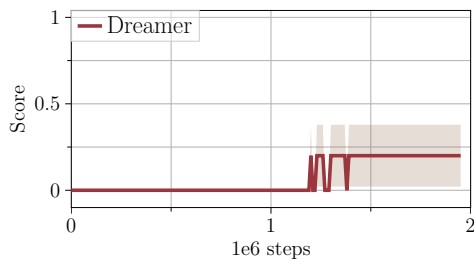

*Figure 22.* **Dreamer in Robodesk** for `upright_block_off_table` (5 seeds, ± SEM).

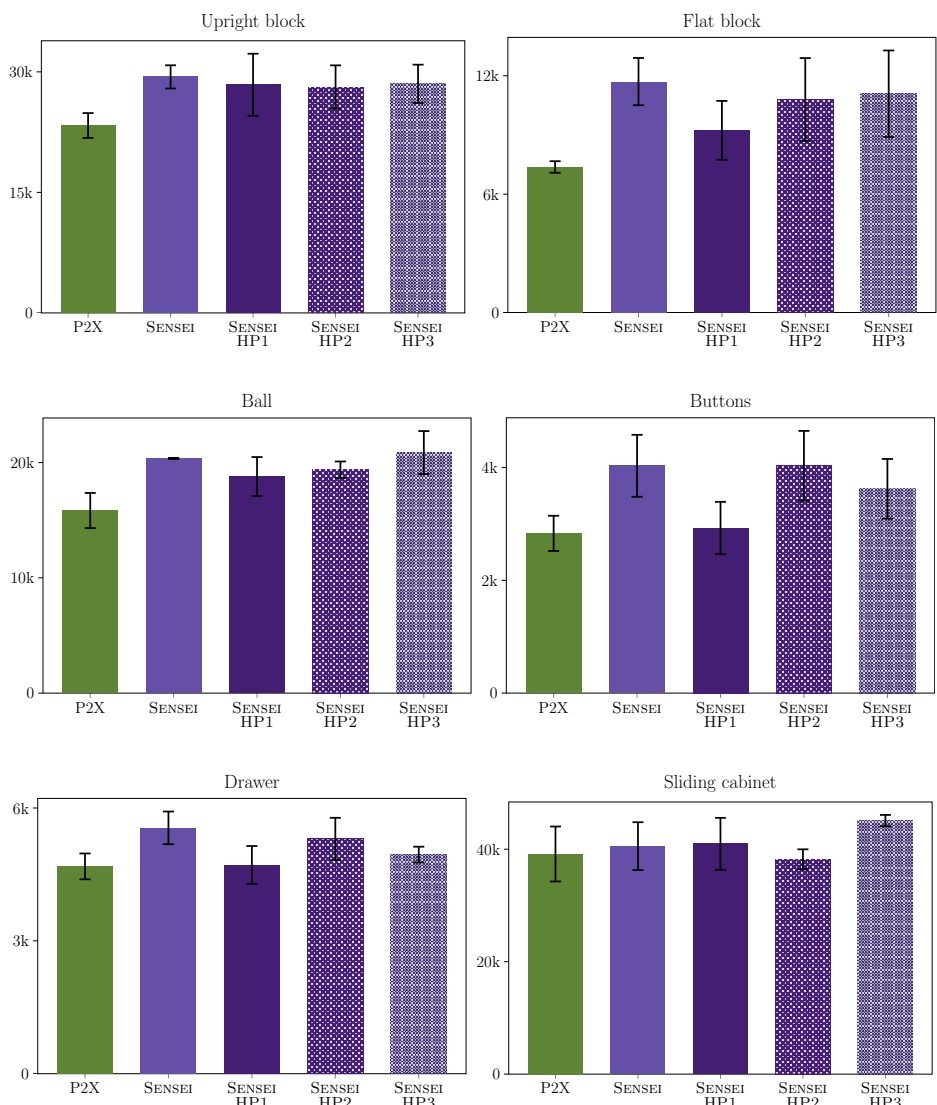

*Figure 20.* **Comparing interactions in Robodesk for SENSEI with different hyperparameters**: We plot the mean over the number of interactions with any object during 1M steps of exploration for SENSEI (winner hyperparameter configuration) and Plan2Explore (P2X) and SENSEI with different hyperparameters as specified in Table 1. Error bars show the standard deviation (3 seeds).

Fig. 21 shows the performance of task-specific policies over training. The world model explored by SENSEI enables significantly faster learning compared to Plan2Explore. SENSEI reliably learns to open the drawer and push the block off the table, though it does not fully converge to 100% success on the ball-lifting task. Nevertheless, across all tasks, SENSEI achieves higher success rates than Plan2Explore.

Finally, we also train a DreamerV3 agent for one of the tasks (`Upright-Block-Off-Table`) from scratch (Fig. 22). Dreamer does not fully learn to solve the task within 2M environment steps. This shows that exploration is crucial to reliably learn to solve these sparse reward tasks.

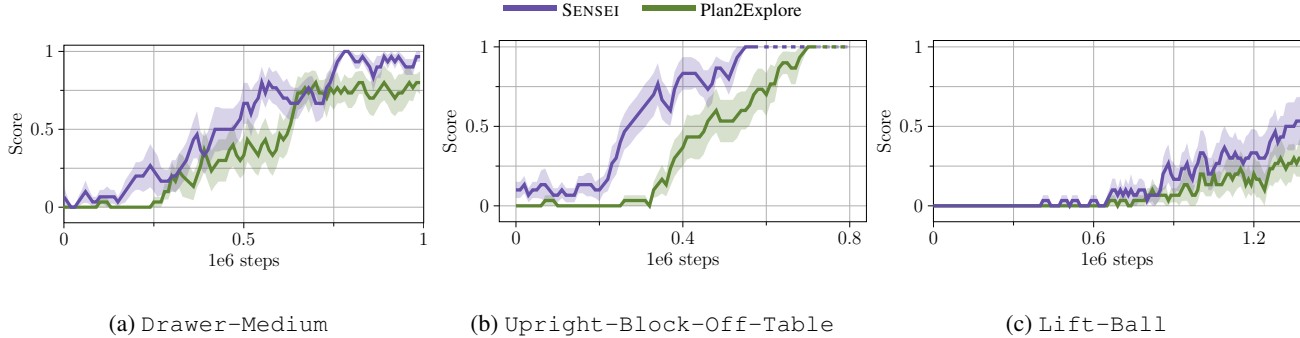

(a) `Drawer-Medium`     (b) `Upright-Block-Off-Table`     (c) `Lift-Ball`

*Figure 21.* **Downstream task performance in Robodesk**: We plot the mean of the episode score obtained during evaluation for three exemplary Robodesk tasks (**a**) `Drawer-Medium`, (**b**) `Upright-Block-Off-Table`, and (**c**) `Lift-Ball`, with world models learned from SENSEI vs. Plan2Explore (P2X) exploration. Shaded areas depict the standard error (10 seeds) and we apply smoothing over the score trajectories with window size 3.

### E.8. Pokémon Red: Extended Results

We evaluate the exploration of battle-related mechanics in Pokémon Red, by tracking the Pokémon party size and their levels and plot their distribution in Fig. 23. Only Plan2Explore and SENSEI explored the battle mechanics of catching Pokémon and leveling them. Dreamer fails to consistently engage in battles or catch Pokémon. Throughout exploration SENSEI assembles the strongest teams in terms of highest individual level and summed total levels of Pokémon in the party (Fig. 23b & 23c).

We visualize how exploration progressess over episodes in Fig. 24. Despite being only trained on task rewards, Dreamer is quickly outperformed by SENSEI and Plan2Explore in terms of rewards achieved (Fig. 24a). This highlights the complexity of the Pokémon environment and how easily agents can get stuck in local optima without structured exploration. Overall, SENSEI and Plan2Explore reach comparable reward levels.

How do SENSEI and Plan2Explore allocate their exploration? Plan2Explore appears to prioritize catching Pokémon: after around 50 episodes, it consistently maintains a larger party than SENSEI (Fig. 24b). This is also reflected in the number of Pokémon caught by both methods (Fig. 25). Plan2Explore manages to collect a wider variety of Pokémon across seeds. SENSEI, on the other hand, allocates more resources to map-based progress towards the Gym (Fig. 24c) and to leveling its Pokémon team through battles, consistently reaching higher Pokémon levels from episode 100 onward (Fig. 8c).

We hypothesize that SENSEI 's exploration is shaped by GPT-4's preferences and our prompt emphasizing the goal of defeating the first Gym Leader. Most early-game wild Pokémon are weak against Brock's Rock/Ground-type team, and only the Water-type starter Squirtle has a type advantage. As a result, the most promising strategy, reflected in SENSEI 's behavior, is to repeatedly battle and level Squirtle in preparation for the Gym.

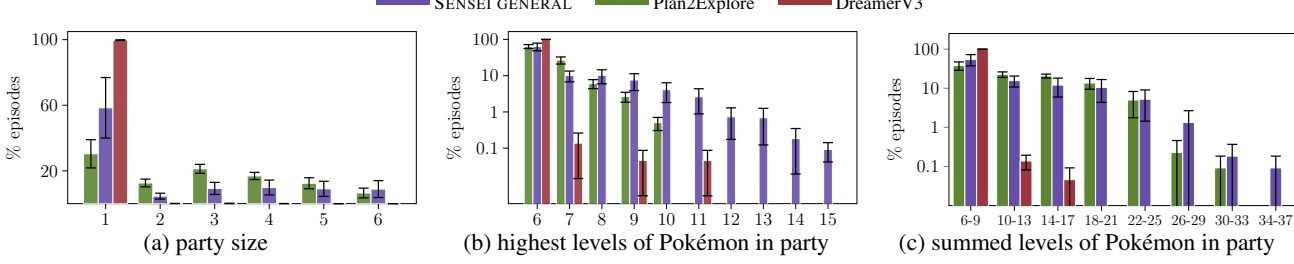

(a) party size     (b) highest levels of Pokémon in party     (c) summed levels of Pokémon in party

*Figure 23.* **Pokémon interaction statistics**: We report the distribution of maximum party size (**a**), the highest individual Pokémon level (**b**), and the total sum of levels across all Pokémon in the party (**c**) for 750k steps of SENSEI, Plan2Explore, and DreamerV3. Error bars indicate standard error across 5 seeds.

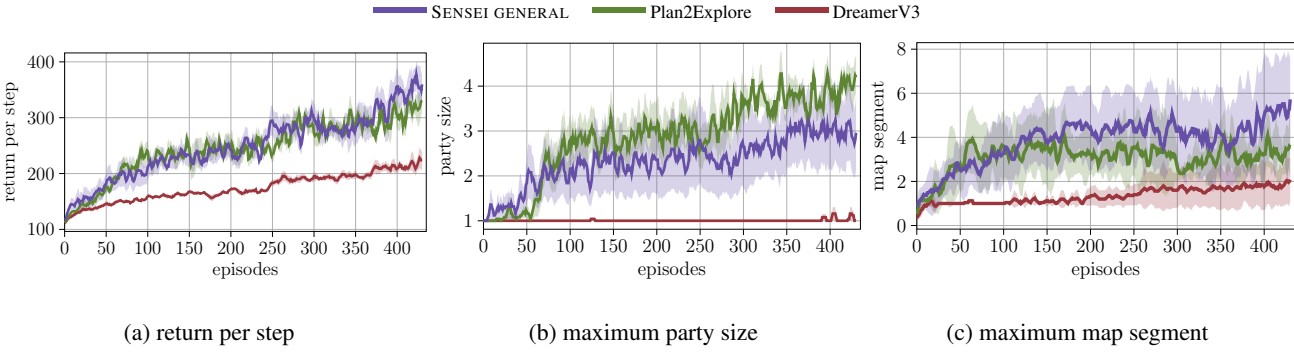

(a) return per step  (b) maximum party size  (c) maximum map segment

*Figure 24.* **Exploration progress in Pokémon Red**: We plot the mean return per step (**a**) and maximum party size reached (**b**) over episodes. To visualize map progress, we plot the maximum map segment reached between episode starting point (0) and the Gym (9) (**c**). We compare SENSEI GENERAL to Plan2Explore ($r_t + \alpha r_t^{\text{dis}}$) and Dreamer ($r_t$). Shaded areas depict the standard error (5 seeds) and we apply smoothing over the score trajectories with window size 5.

### E.9. Pokémon Red: Second Generation VLM-MOTIF

A limitation of our current SENSEI implementation is that if an agent encounters entirely novel states, the semantic reward may contain mostly noise, as it is outside of the training distribution for VLM-MOTIF. Here, exploration relies on epistemic uncertainty rewards ($r_t^{\text{dis}}$), similar to Plan2Explore. This can be the case for large, open-ended environment such as in Pokémon Red. The initial SENSEI dataset contains 100K pairs from a Plan2Explore run (500K steps exploration), reaching at most Viridian Forest (map segment 6, Fig. 8a). SENSEI reaches the frontier of Viridian Forest more often, occasionally breaking into Pewter City, where the first Pokémon Gym is located. However, the first-generation VLM-MOTIF has never seen this area and thus relies on information gain to explore beyond Viridian Forest.

We visualize these failure cases in Fig. 26. For instance, the first-generation reward function incorrectly assigns higher interestingness to a task-irrelevant museum (top row, frames 3 & 5) than to the Pewter Gym (top row, frames 2 & 4), which is the actual task goal. Upon entering the Gym, the semantic reward drops sharply (middle row, frames 3–5), further highlighting this OOD failure.

To address this, we propose a simple refinement procedure: re-annotating data collected from a SENSEI run. We sample 50K additional observation pairs explored by SENSEI and annotate them using the same VLM prompting strategy. We then distill a second-generation semantic reward function $R_\psi$, trained on both the original and newly collected annotations, now including previously unseen areas such as Pewter City and the Gym. As shown qualitatively in Fig. 26 (Generation 2), the refined reward function corrects earlier misjudgments: it spikes correctly upon seeing the Gym and peaks when the agent enters and faces the Gym Leader (middle row, frame 5). This demonstrates the potential of iterative semantic refinement. We see this as a promising extension to SENSEI, enabling agents to continuously improve their internal reward models by incorporating data from newly explored regions.

We evaluate the effect of the refined semantic reward function on exploration by continuing the original SENSEI runs for 200 additional episodes. Specifically, we resume exploration from the same point used to collect the second-generation annotations, comparing runs that use either the refined reward function (Generation 2) or the original (Generation 1). To

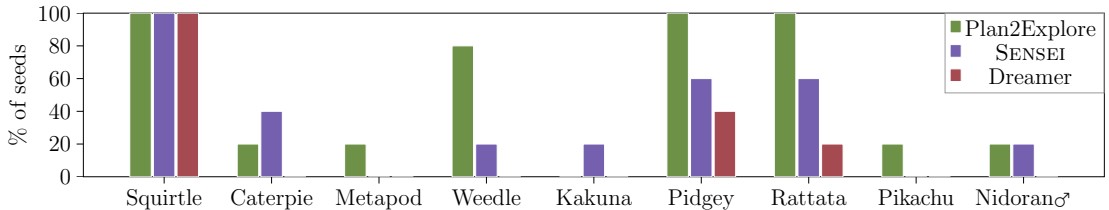

*Figure 25.* **Pokémon caught**: For SENSEI GENERAL, Plan2Explore and DreamerV3 we plot the ratio of seeds that managed to obtain the listed Pokémon at least once during 750k steps of exploration. Other Pokémon were never caught.

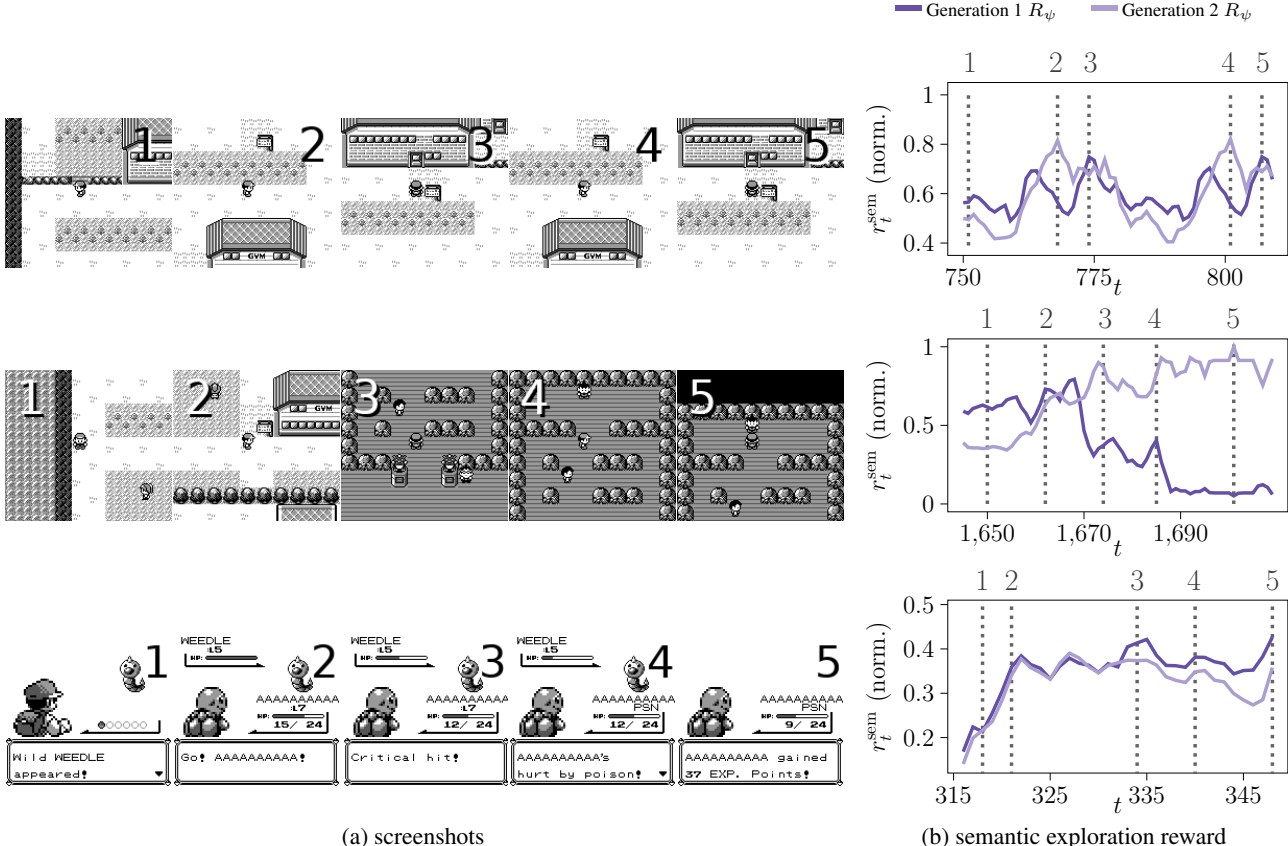

(a) screenshots                                                    (b) semantic exploration reward

*Figure 26.* **First and second generation semantic rewards in Pokémon Red**: We visualize semantic rewards from a reward function $R_\psi$ learned either purely from Plan2Explore data (Generation 1) or refined based on SENSEI data. Generation 2 semantic rewards $r_t^{sem}$ correctly peak when seeing Pewter Gym (top row, 2 & 4) or entering the Gym (middle row, 3 –5). These images are out-of-distribution for Generation 1 $R_\psi$, which is incorrectly yielding low sematic rewards $r_t^{sem}$. In battle-related game play both reward functions are highly correlated.

assess whether simply running our baselines longer would yield similar benefits, we also continue one run of Plan2Explore. For fairness and to avoid biasing our baselines, we select the only Plan2Explore seed that managed to reach Pewter City.

Exploration progress across these runs is visualized in Fig. 27, where a dashed gray line marks the onset of the second-generation annotations and the divergence point between Generation 1 and Generation 2. SENSEI Generation 2 broadly continues the exploration trends of Generation 1, outperforming Plan2Explore on all metrics except Pokémon party size (Fig. 27e). However there are a few subtle differences between SENSEI Generation 1 & 2: Both SENSEI Generation 1 & 2 seem to mostly avoid catching new Pokémon (Fig. 27e) and instead focus solely on leveling up their Squirtle. Thereby, SENSEI Generation 2 reaches slightly higher levels than Generation 1 (Fig. 27b). We hypothesize that SENSEI's focus on leveling Squirtle comes from our prompt defining its objective "to find and defeat the Gym Leader, Brock". The best option to beat Brock is a high-level Squirtle whose Water attacks are super effective against Brock's team. While SENSEI Generation 2 visits fewer overall map segments than Generation 1 (Fig. 27d), it concentrates more on segments 8 and 9 (Fig. 27c), corresponding to Pewter City and the Pewter Gym. This focused exploration, combined with a stronger Squirtle, allows SENSEI Generation 2 to achieve a critical milestone: defeating Brock, the Pewter City Gym Leader, and obtaining the Boulder Badge. This event occurs twice over the course of continued exploration (Fig. 27f), with the first badge earned after 570 episodes, or roughly 1.8 million environment steps. None of our baselines were able to achieve this milestone before.

This demonstrates the potential of iterative semantic refinement. We see this as a promising extension to SENSEI, enabling agents to continuously improve their internal reward models by incorporating data from previously unexplored regions.

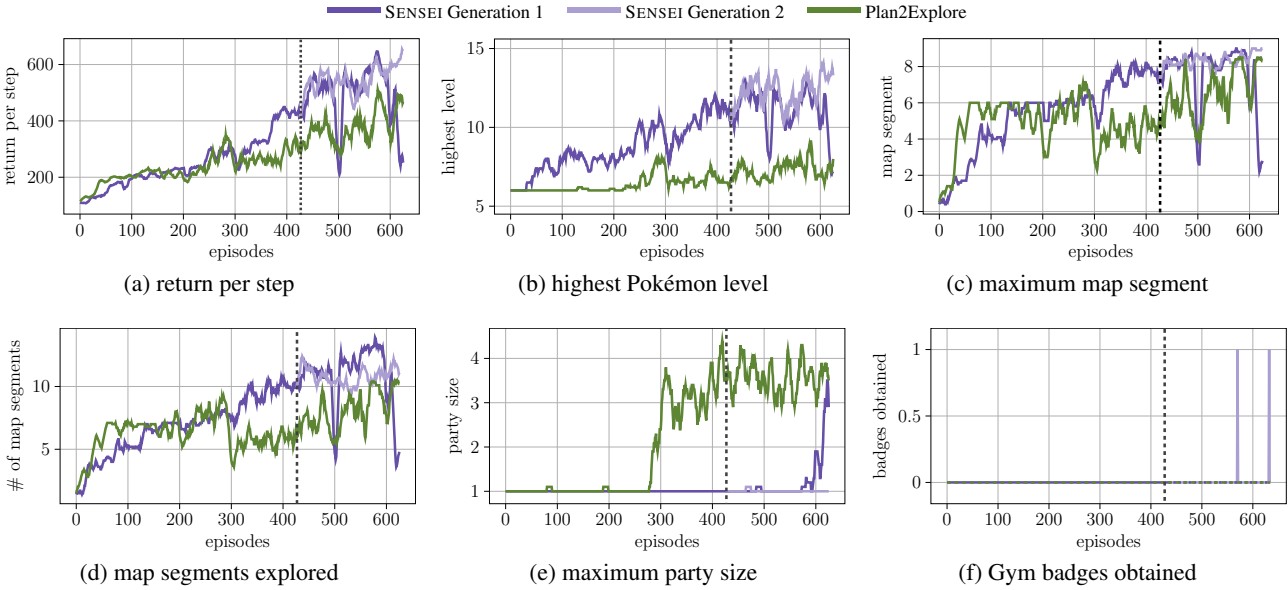

*Figure 27.* **Exploration with first and second generation semantic rewards in Pokémon Red**: We track exploration progress when continuing one seed of SENSEI with $R_\psi$ annotated on its previous exploration data (Generation 2) and compare against continuing with the previous $R_\psi$ (Generation 1) or longer runs of Plan2Explore. We plot return per step (**a**), highest Pokémon level achieved (**b**), maximum map segment reached (**c**), overall number of map segments explored (**d**), maximum party size reached (**e**) or Gym badges obtained (**f**) over episodes. The gray dashed line marks the annotation Generation 2 $R_\psi$. In (**a**)–(**e**) we apply smoothing over the trajectories with window size 10.

## F. Computation

SENSEI has 3 phases: (1) annotation of data pairs (offline), (2) reward model, i.e. VLM-MOTIF, training (offline), (3) online RL training with environment interactions (DreamerV3). All experiments were performed on an internal compute cluster.

**Dataset Annotation**    The annotation of data pairs is done using the OpenAI API, such that a single CPU is sufficient. For instance for Robodesk with a dataset size of 200K pairs, we parallelized this over 200 CPUs, where we annotated 1K pairs each, which took approximately 40 minutes. Note that annotations are fully offline and do not affect the runtime of SENSEI itself. Each annotation using the single-turn strategy cost $0.002 with `gpt-4o-2024-05-13` and $0.004 with `gpt-4-turbo-2024-04-09`. The multi-turn prompting for the zero-knowledge Robodesk annotations also cost $0.004 per pair with `gpt-4o-2024-05-13`.

**Reward Model Training**    After annotating the dataset, we train the VLM-MOTIF network using a single GPU for 50 epochs. Using e.g. Tesla V100-SXM2-32GB, this took 20min. We ran a grid search over different hyperparameters for VLM-MOTIF training (batch size, learning rate, weight decay, network size), testing for a total of 18 different combinations, and we chose the reward model with the best validation loss to use in SENSEI runs.

**Online Model-based RL Training**    SENSEI is built on top of DreamerV3, just like our main baseline Plan2Explore. On a NVIDIA A100-SXM4-80GB, SENSEI runs at ca. 7.5Hz, Plan2Explore runs at ca. 10Hz and pure VLM-MOTIF runs at ca. 8.7Hz.

