# OpenReview forum: "SENSEI: Semantic Exploration Guided by Foundation Models to Learn Versatile World Models"
_ICML.cc/2025/Conference — ICML 2025 poster_

### Official Review · Reviewer_bmzW · 2025-02-24

**Overall Recommendation:** 3

**Summary:**

This paper introduces SENSEI (SEmaNtically Sensible ExploratIon), a LM-based framework for guiding the exploration phase of RL agents towards "interesting" states. It trains a reward model based on VLM-generated rankings of interestingness on prior exploration data. The final exploration reward is the sum of the aforementioned interestingness score and an uncertainty score computed by ensemble disagreement (taken from Plan2Explore [1]). It dynamically reweights these two scores to first go towards an interesting state, then explore from there. Experiments are performed in two domains: MiniHack (navigating dungeons and finding keys) and Robodesk (simulated robot arm interacts with objects on a desk). Results show that this method, compared to baselines:
1. succeeds in directing towards the agent towards semantically meaningful states during exploration
2. solves tasks faster

**Claims And Evidence:**

Examining the evidence presented for the main claims made in the submission (1 & 2 in summary):
1. SENSEI guides LMs towards "interesting" states: The evidence shown for this claim includes a small amount of qualitative samples (Figure 4), and a quantitive analysis showing evidence that agents generally interact more with objects under SENSEI than P2X. The precie definition of "interestingness" is worth some discussion in the paper: For example, why are object interactions considered "interesting"? Breakdown of other types of interactions deemed interesting by the reward model could be useful: e.g. seeing information-dense states. Are different types of interactions on the same object considered more or less interesting (e.g. pressing a button vs. moving it)? Furthermore, can interestingness be adjustable based on existing information (e.g. a new object is more interesting than an object seen before)? This claim can benefit from being slightly more precise -- or including a more rigorous analyses of what types of states have high "interesting" reward.
2. SENSEI solves task faster: This seems fairly well-substantiated from the end-to-end experiments in Figure 4. SENSEI outperforms Plan2Explore, DreamerV3, and PPO, all of which do not use a interestingness-based reward. It could be clarified that P2E is basically exactly the same as SENSEI but without the interesting-ness reward, representing a minimal ablation of the contributed portion.

**Essential References Not Discussed:**

None

**Experimental Designs Or Analyses:**

The experimental designs generally seem sound. Experiments were conducted on two different (simulated) environments. Comparisons against Plan2Explore (the method upon which this paper is based -- which is missing the interestingness score) and Random Network Distillation (a model-free method trained with PPO), demonstrate 1) evidence of increased semantically-meaningful object interactions, 2) empirical improvements on downstream tasks, when the interestingness score is incorporated into the model.

One question is how much these environments may already be in GPT-4o, which may already a sense of what goals exist in these environments, and thus output "interestingness" scores based on proximity to the goal. There is also a question of how large and diverse these environments are, whereby the goal state may be the only "interesting" state there is.

Some additional analyses to consider:
1. Is downstream performance improvement predicated on the goal state (or a subgoal state) being one of (or proximate to) the identified interesting states? What happens in settings where goals may not necessarily be aligned with "interestingness"?
2. To what extent is GPT4-o necessary for ranking "interestingness" and not just heuristics like total information content of the state?
3. To what extent is GPT4-o's rating of interestingness generalizable to diverse (potentially real-world) environments and goals?

**Methods And Evaluation Criteria:**

SENSEI relies on data that is pre-collected from an exploration policy in order to distill an "interesting-ness" reward function from. Given that the policy relies on pre-collected data, it seems like an unfair comparison to compare against baselines that are purely from-scratch and not bootstrapped with any initial exploration data: perhaps a fairer comparison (and a more realistic setting) would be to look at the transferability of the distilled interesting-ness model across environments (even though this itself would mean SENSEI has access to some extra domain-specific information that other methods don't), or start P2X with the world model it's built up with the existing exploration data, and have it continue exploring from there. Why not just use the VLM online to give interestingness rewards?

The choice of evaluation datasets makes sense to me. The paper shows improvements on two very different domains: one with navigating a dungeon and one with interacting with objects on a desk. Perhaps adding a real-world domain would also be useful here (but I don't think it's essential for this work).

**Other Comments Or Suggestions:**

None

**Other Strengths And Weaknesses:**

Overall, the paper is quite comprehensive with its experiments/analyses and has done a good job making a case for the necessity of an interestingness score. The two phases of exploration seems like a very interesting concept.
My main concerns have been listed above: specifically, I think the two critical concerns I have with this paper are:

1. Novelty with respect to prior literature: there has been lots of prior work on LLM/VLM-guided exploration, and it is unclear why SENSEI is advantageous over these prior methods. It would be great if the authors to include these baselines, or at least a discussion of when/why SENSEI is better than them.

2. The method assumes access to a *pre-collected* dataset of exploration data on the specific environment that the agent is trying to explore. It is unclear to me when this setting where pop up in the real world, and why this data isn't just used to initialize the agent, who can then choose to continue exploring from there (if it so desires). Moreover, none of the baselines assume access to this pre-collected dataset, so the evaluation isn't quite comparable.

**Questions For Authors:**

If you have any important questions for the authors, please carefully formulate them here. Please reserve your questions for cases where the response would likely change your evaluation of the paper, clarify a point in the paper that you found confusing, or address a critical limitation you identified. Please number your questions so authors can easily refer to them in the response, and explain how possible responses would change your evaluation of the paper.

1. it is unclear to me why the distillation step is even necessary: why not just use the VLM directly online?
2. it is unclear whether the VLM is even necessary, or what exactly it is doing -- it would be insightful to compare to heuristic baselines like information content of the image, number of objects in the state, etc.
3. What kinds of states are "interesting" and how is this distinct from (or perhaps the same as) generating human-like subgoals from earlier papers?

Please also see questions from above. More details about baselines Plan2Explore and RND would be useful in the main paper.

**Relation To Broader Scientific Literature:**

Using LLMs to provide semantic signal for exploration has been studied extensively over the past few years, including by certain methods cited by the authors themselves (e.g. ELLM, OMNI, LAMP). It's unclear to me why this method would be preferred over others. For example, ELLM does not require exploration data to be gathered ahead of time, so might even be preferable in certain cases. (On the other hand, if the distilled model is general across environments, I can see a cost-based argument for not needing to re-query GPT-4o each time a new environment is encountered -- though I am not certain that is the case for this method.) A comparison to these methods should be included as baseline(s), or at the very least a discussion on when/why would SENSEI be preferred over these techniques.

**Theoretical Claims:**

No theoretical claims in paper

---

> ### Author Rebuttal · Authors · 2025-04-01
>
> We thank the reviewer for the constructive review. We appreciate that you found our paper comprehensive, the introduced concepts interesting, and our experimental design sound.
>
> ### VLM-based exploration baselines
>
> Existing VLM-based exploration methods (Omni, Motif, ELLM) rely on assumptions that limit their applicability to our setting, such as requiring expert demonstrations (Motif), language-grounded environments (Omni, Motif, ELLM), or high-level actions (Omni, ELLM). In contrast, SENSEI operates in a more general RL setup without these constraints.
>
> Motif also uses an initial offline expert dataset. Omni requires access to the full task list and corresponding rewards. Motif and ELLM both rely on event-counting to bias novelty. In contrast, we show how novelty and semantic interestingness can be seamlessly combined with minimal assumptions, enabling deployment in low-level control environments while preserving scalability.
>
> Additionally, none of these methods use model-based RL, which is central to SENSEI’s sample-efficient exploration. For fairness, we adapt Motif into our VLM-Motif baseline. Our results show SENSEI outperforms VLM-Motif, highlighting the benefits of dynamic reweighting and the importance of incorporating information gain within a model-based approach. We will add this discussion to the final version.
>
> ### Offline VLM usage
>
> For details on why offline VLM usage is necessary/preferred and on our distillation procedure, please see our answer to Reviewer 7mHN (Section: Approximating human interestingness).
>
> ### Diversity of Envs and corresponding interestingness
>
> > There is also a question of how large and diverse these environments are, whereby the goal state may be the only "interesting" state there is.
>
> We now include a diverse environment: Pokémon Red. It requires navigating a large map and mastering battle mechanics to catch/fight wild Pokémon and defeat Trainers. Here, interestingness is multi-faceted (map progression, Pokémon collection, etc.), and SENSEI shows consistent gains. Please see our response to Reviewer 7mHN and our rebuttal webpage: [https://sites.google.com/view/sensei-icml-rebuttal](https://sites.google.com/view/sensei-icml-rebuttal).
>
> ### Pre-Collected Dataset
>
> SENSEI uses pre-collected data, unlike Plan2Explore, but our dataset size is chosen to ensure coverage of interesting states—not to be minimal. Crucially, SENSEI remains effective with significantly less data. In a new ablation, we show that in MiniHack-KeyRoom, using just ¼ of the dataset (25k samples) yields similar interaction statistics (Webpage Figure E5).
>
> In real-world settings, various offline robotics datasets (e.g., OpenX-Embodiment, TriFinger RL) contain expert and non-expert control data, making SENSEI directly applicable. Alternatively, SENSEI could begin with exploration purely via $r^{dis}$, then distill the reward from collected data. We appreciate this suggestion and will include the discussion in the final version. It aligns well with our generational SENSEI perspective, further supported by the Pokémon experiment (see response to Reviewer 9xtC).
>
> ### Alignment Between Goals and "Interestingness"
>
> If a task’s goals or subgoals don’t align with "interestingness", SENSEI relies on epistemic uncertainty ($r^{dis}$) to discover them. In practice, this has not been limiting.
>
> For instance, in MiniHack-KeyRoom, the goal is to reach a staircase (Fig. 4, top, img 5), which GPT-4 does not consider especially interesting. Standing near the exit yields lower semantic rewards (point 5) than standing near the key and door (points 3&4). Still, SENSEI reliably solves the task and collects more rewards than Plan2Explore by reaching the staircase.
>
> ### Interestingness vs Information content
>
> Interestingness captures semantic relationships between entities, which may not be reflected in simple information content heuristics.
>
> For example, in MiniHack-KeyChest, compare: (1) “Agent is next to a locked chest with a key in the inventory” and (2) “Agent is next to a locked chest with no key”. Both have similar information content, but (1) is more interesting due to the key-chest relation.
>
> > Can interestingness be adjusted based on existing information (e.g., a new object is more interesting than a familiar one)?
>
> This is precisely what SENSEI achieves by combining interestingness with disagreement (information gain). We aim to explore regions that are both interesting and novel.
>
> > What kinds of states are "interesting"? How is this distinct from human-like subgoals in earlier work?
>
> We assume VLMs incorporate priors aligned with human preferences. Thus, human-like subgoals would likely be favored by VLMs over random states. However, unlike prior goal-based work, we don’t assume access to a subgoal set. Instead, we use Motif to compute a continuous reward, approximating interestingness.
>
> We hope this response addresses the reviewer’s concerns and we would be happy to clarify further if needed.

---

### Official Review · Reviewer_wRRZ · 2025-02-27

**Overall Recommendation:** 3

**Summary:**

The paper introduces a novel framework for intrinsic motivation in reinforcement learning (RL) agents, enabling them to explore environments meaningfully without relying on task-specific rewards. The authors propose SEmaNtically Sensible ExploratIon (SENSEI), which leverages Vision Language Models (VLMs) to guide exploration by distilling a reward signal reflecting observations' semantic interestingness. The method demonstrates the ability to discover meaningful behaviors in complex environments and promises to accelerate downstream task learning.

**Claims And Evidence:**

The claims in the SENSEI paper are supported mainly by clear and convincing evidence, though some aspects could benefit from further validation or broader testing.

The paper suggests SENSEI could scale to real-world applications, but experiments are limited to simulated environments (MiniHack, Robodesk). Photorealism and occlusion handling are acknowledged limitations.

The paper lacks comparisons to recent exploration methods beyond P2X (e.g., Curiosity-Driven Exploration via Disagreement, Skew-Fit). This limits the strength of claims about SENSEI’s superiority.

**Essential References Not Discussed:**

In most real-world scenarios, there are potential complex disturbances that can affect the model's learning and judgment. Task scenarios like Noisy TV [1,2,3] may lead to excessive exploration of novel areas, resulting in exploration failure.

**References**

[1] urgen Schmidhuber, J. (1991). Adaptive confidence and adaptive curiosity. Technical Report FKI {149 {91 (revised), Technische Universit at M unchen, Institut f ur Informatik.

[2] Mavor-Parker, A., Young, K., Barry, C., & Griffin, L. (2022, June). How to stay curious while avoiding noisy tvs using aleatoric uncertainty estimation. In International Conference on Machine Learning (pp. 15220-15240). PMLR.

[3] Huang, K., Wan, S., Shao, M., Sun, H. H., Gan, L., Feng, S., & Zhan, D. C. (2025). Leveraging Separated World Model for Exploration in Visually Distracted Environments. Advances in Neural Information Processing Systems, 37, 82350-82374.

**Experimental Designs Or Analyses:**

Yes. The experimental designs and analyses in the SENSEI paper are mostly sound, with clear motivations and appropriate metrics. However, the robustness of VLM annotations, the quality of the world model, and the diversity of tasks could be more rigorously tested. Addressing these issues would strengthen the paper’s claims and provide a more comprehensive evaluation of the proposed methods.

**Methods And Evaluation Criteria:**

The SENSEI paper's proposed methods and evaluation criteria are well-aligned with the problem of semantically guided exploration in reinforcement learning (RL). There are some potential improvements for this paper: (1) Including more recent exploration methods (e.g., Curiosity-Driven Exploration via Disagreement, Skew-Fit) would strengthen the evaluation by providing a broader context for SENSEI’s performance. (2) Evaluating SENSEI on a wider range of tasks (e.g., navigation, manipulation, and social interaction) could further demonstrate its versatility.

**Other Comments Or Suggestions:**

See questions.

**Other Strengths And Weaknesses:**

**Strengths**

1. The paper is well-organized, clearly explaining the method, experiments, and results.
2. The experiments are thorough, with ablation studies (e.g., Figure 17 for fixed reward weighting, Figure 18 for hyperparameter sensitivity) and comparisons to strong baselines (e.g., Plan2Explore, RND). The results are presented with standard errors and multiple seeds, ensuring reproducibility.

**Weaknesses**

1. The paper lacks comparisons to state-of-the-art exploration methods like Curiosity-Driven Exploration via Disagreement or Skew-Fit. Including these would provide a more comprehensive evaluation of SENSEI’s performance.
2. The evaluation focuses on two MiniHack tasks and a subset of Robodesk tasks. Testing SENSEI on a broader range of tasks (e.g., navigation, social interaction) would better demonstrate its versatility.
3. This paper lacks a detailed methodological comparison with Plan2Explore. However, if viewed from another perspective, it resembles an exploration of the intrinsic rewards of disagreement based on the world model's ensemble dynamics in Plan2Explore, adding a VLMs reward related to the task's interestingness, and then considering how to balance these two rewards. Of course, engineering improvements are also important, especially if the results are outstanding.

**Questions For Authors:**

1. In Figure 8, why do you choose observations from view (b) apart from the reasons you mentioned in the paper? Have you conducted any ablation studies on this design?
2. From the provided prompts, the rewards given by the large model are based on interestingness. However, the definition of interestingness is a very subjective concept. Is the exploration of all tasks considered interesting?
3. In Figure 7, SENSEI has already trained for 500K steps before fine-tuning on downstream tasks. Would it be fair to let DreamerV3 train for an additional 500K steps before conducting a formal comparison of downstream tasks?
4. In most real-world scenarios, there are potential complex disturbances that can affect the model's learning and judgment. Task scenarios like Noisy TV [1,2,3] may lead to excessive exploration of novel areas, resulting in exploration failure. Have you considered these noisy exploration scenarios?

**References**

[1] urgen Schmidhuber, J. (1991). Adaptive confidence and adaptive curiosity. Technical Report FKI {149 {91 (revised), Technische Universit at M unchen, Institut f ur Informatik.

[2] Mavor-Parker, A., Young, K., Barry, C., & Griffin, L. (2022, June). How to stay curious while avoiding noisy tvs using aleatoric uncertainty estimation. In International Conference on Machine Learning (pp. 15220-15240). PMLR.

[3] Huang, K., Wan, S., Shao, M., Sun, H. H., Gan, L., Feng, S., & Zhan, D. C. (2025). Leveraging Separated World Model for Exploration in Visually Distracted Environments. Advances in Neural Information Processing Systems, 37, 82350-82374.

**Relation To Broader Scientific Literature:**

The SENSEI paper's key contributions are closely related to the broader scientific literature on reinforcement learning (RL), intrinsic motivation, and the use of foundation models for exploration.

Intrinsic Motivation: SENSEI builds on intrinsic motivation in RL, where agents explore environments without external rewards. Prior methods (e.g., curiosity-driven exploration, prediction error, Bayesian surprise) focus on low-level interactions, while SENSEI leverages VLMs to guide exploration toward semantically meaningful behaviors (similar to children’s play).

Model-Based RL: SENSEI uses a Recurrent State Space Model (RSSM) to predict semantic rewards, aligning with prior work on world models (Ha & Schmidhuber, 2018; Hafner et al., 2023). The key innovation is predicting semantic rewards directly from latent states, enabling the agent to evaluate hypothetical states.

**Theoretical Claims:**

There is no proof in the paper.

---

> ### Author Rebuttal · Authors · 2025-04-01
>
> We thank the reviewer for their feedback and greatly appreciate that they found our paper well-organized, our experiments thorough, and our evidence clear and convincing. We aim to address the remaining concerns below.
> > The paper lacks comparisons to recent exploration methods beyond P2X like Curiosity-Driven Exploration via Disagreement or Skew-Fit
>
> First, we note that we compare not only against P2X, but also RND and VLM-Motif—a reimplementation of Motif adapted to our world model and VLM setting. Thus, we compare against 3 strong exploration baselines.
>
> Second, Plan2Explore represents the state-of-the-art in curiosity-driven exploration via disagreement for pixel-based inputs. We are not aware of other disagreement-based methods suitable for comparison.
>
> Skew-Fit, on the other hand, falls into the goal-based exploration paradigm, where goals are sampled to maximize state coverage. Our RND baseline also targets state-space coverage. We believe goal-based methods are orthogonal to our setting. In addition, we note that Hu el al. 2023 [1] shows (Fig. 4 of main paper) that an improved model-based adaptation of Skew-Fit is performing worse or similar to Plan2Explore on a range of environments, making Plan2Explore the stronger baseline to compare to.
>
> > Testing SENSEI on a broader range of tasks (e.g., navigation, social interaction) would better demonstrate its versatility.
>
> We have now added a new environment—Pokémon Red—which involves navigation over a large map and interactions with Pokémon and gym leaders. See our response to Reviewer 7mHN for details and our rebuttal webpage (https://sites.google.com/view/sensei-icml-rebuttal) for results.
>
> > This paper lacks a detailed methodological comparison with Plan2Explore.
>
> Our method builds on Plan2Explore by adding semantic rewards and dynamic reward scaling. We apologize if this was unclear. We have clarified this connection in the method section and discussion.
> We also empirically analyze the contributions of both additions via comparisons to VLM-Motif and through ablations on dynamic reward scaling (Suppl. D.6).
>
> > In Figure 8, why do you choose observations from view (b) apart from the reasons you mentioned in the paper?
>
> In Robodesk’s default view, the robot arm often occludes objects, presenting challenges for learning. Early tests also showed better VLM annotations from the right camera view, which we therefore used.
>
> > However, the definition of interestingness is a very subjective concept. Is the exploration of all tasks considered interesting?
>
> We agree that interestingness is subjective. However, we argue that across cultures, commonalities exist in what humans find interesting—reflected in the large-scale training data of VLMs. The tasks in our environments were all designed by humans, implying inherent human interest.
>
> For annotation with SENSEI-General in Robodesk, we first ask the VLM for a description of the environment and what could be interesting. We observe strong alignment between the VLM’s responses and the environment’s task distribution (Supp. C.3.2).
>
> > In Figure 7, SENSEI has already trained for 500K steps before fine-tuning on downstream tasks. Would it be fair to let DreamerV3 train for an additional 500K steps?
>
> Excellent point—our method does spend more time in the environment. However, note that SENSEI does not optimize for any particular task during exploration. Even with additional training, Dreamer, in our experiments, does not match SENSEI’s performance.
>
> For example, in MiniHack-KeyRoomS15, SENSEI solves the task reliably after 300K steps of task-based training (on top of 500K steps exploration), totaling 800K steps. Dreamer, even after 1M steps, does not reliably solve the task across seeds.
>
> To emphasize this further, we now compare SENSEI in a Robodesk task (Upright-Block-Off-Table) after 1M exploration steps to Dreamer trained from scratch. SENSEI solves the task across all seeds in 600K steps; Dreamer only succeeds in one seed after 2M steps. Even counting SENSEI’s 1M exploration, it outperforms Dreamer with 1.6M total steps (see our rebuttal webpage, Figure E4).
>
> > Task scenarios like Noisy TV may lead to excessive exploration of novel areas, resulting in exploration failure. Have you considered these?
>
> Indeed, this is a known issue in intrinsically motivated RL. However, due to the ensemble disagreement formulation—used in Plan2Explore and our method—this is not a concern. In stochastic environments, the ensemble predictions converge to the process mean with enough samples, resulting in 0 disagreement and no further reward. We refer the reviewer to [2] for more details.
>
> [1] Hu, E. et al., Planning goals for exploration, ICLR 2023.
>
> [2] Pathak, D. et al., Self-supervised exploration via disagreement, ICML 2019.

---

> > ### Comment · Reviewer_wRRZ · 2025-04-08
> >
> > Thanks for the author's reply. My problems have been solved, and I will keep the score unchanged.

---

### Official Review · Reviewer_9xtC · 2025-03-13

**Overall Recommendation:** 3

**Summary:**

The paper proposes SENSEI, a framework designed to enhance exploration in model-based RL by integrating semantic guidance from VLMs. SENSEI distills a reward signal of interestingness from VLM-generated annotations of observations, guiding agents toward semantically meaningful interactions. This intrinsic reward system enables the learning of versatile world models, with internal models of interestingness, improving exploration efficiency. Empirical results in robotic and video game simulations demonstrate that SENSEI successfully encourages agents to discover meaningful behaviors from low-level actions and raw image observations, significantly enhancing downstream task learning.

**Claims And Evidence:**

Yes, the claims are supported by clear and convincing evidence.

**Essential References Not Discussed:**

I don’t have any additional ones to suggest.

**Experimental Designs Or Analyses:**

Yes, the experiments are sound and valid. Experiments are conducted over two domains and thorough analyses are done on both.

I appreciate the SENSEI General experiments on Robodesk, showing that SENSEI is not dependent on the injection of external environment knowledge. However, why is SENSEI General not shown in the MiniHack environment? It will be useful to show it as an ablation in both domains.

**Methods And Evaluation Criteria:**

Yes, the proposed method, e.g., learning an internal model of interestingness via SENSEI, is novel and the evaluation criteria make sense for the problem.

In the Robodesk evaluation, it would be useful to include a human evaluation of the produced behaviors to support the claim that “our semantic exploration reward seems to lead to more meaningful behavior than purely epistemic uncertainty-based exploration.”

**Other Comments Or Suggestions:**

See other sections.

**Other Strengths And Weaknesses:**

Strengths
- The proposed method and idea is novel and interesting. Thorough experiments also show how having an internal model of interestingness in the world model can contribute to downstream tasks.

Weaknesses
- See other sections.

**Questions For Authors:**

1. What happens if exploring a behavior in one area unlocks uncertain behaviors in a previously explored area? For example, suppose the agent has already explored a box in a room but then presses a button at the other end of the room, opening the box. Now, previously explored spaces contain newly unlocked, interesting behaviors. How might SENSEI or future extensions handle this scenario?
2. The current version of SENSEI makes use of a fixed distilled reward function. However, what happens if what’s considered interesting changes over time? For example, in robot manipulation, initially, interacting with objects might be interesting, but once the robot has mastered basic manipulation, the focus of interestingness might shift towards more challenging tasks, such as assembling or stacking objects. How might SENSEI perform in scenarios where the definition of interesting behaviors evolves?
3. Relatedly, since the distilled reward function is not updated during world model training, what if the agent discovers behaviors or states not present in the initial self-supervised exploration data? For instance, consider an agent accidentally falling into a ditch. Since the initial exploration data did not include this environment, the distilled reward function lacks information regarding the interestingness of exploring this new scenario. How might SENSEI cope with this situation?

**Relation To Broader Scientific Literature:**

SENSEI is most related to intrinsically motivated RL. Unlike traditional intrinsic exploration methods, which emphasize novelty or state-space coverage, this work integrates semantic guidance from VLMs to provide more human-aligned, meaningful intrinsic rewards, following recent trends of leveraging foundation models for guiding exploration (e.g., Motif, OMNI). Additionally, by incorporating model-based RL frameworks and world models, the approach advances the capability of RL agents to predict semantic “interestingness,” thus bridging human-defined semantic priors with effective exploration in complex environments.

**Theoretical Claims:**

Yes, they are correct.

The writing in Section 2.3 can make the distinction between r_t_sem and r_t_sem_hat clearer.

---

> ### Author Rebuttal · Authors · 2025-04-01
>
> We sincerely appreciate your thoughtful feedback and recognition of SENSEI’s novelty, thorough evaluations, and strong empirical evidence. Below, we address your key questions and minor comments.
>
> ### SENSEI in newly unlocked areas
>
> > What happens if exploring a behavior in one area unlocks uncertain behaviors in a previously explored area?
>
> As the reviewer correctly noted, we differentiate two cases: 1) the newly unlocked behavior exists in the initial self-exploration dataset, 2) the behavior is entirely novel.
>
> Case 1 is naturally accommodated by SENSEI, as these semantic relationships are reflected in the learned VLM-Motif. For example, in MiniHack-KeyChest, an agent might initially discover a locked chest before finding the key. Upon acquiring the key, the previously explored chest gains new semantic significance, yielding higher semantic rewards. SENSEI adapts seamlessly, as semantic meaning evolves with discoveries. In contrast, Plan2Explore and other uncertainty-driven methods lack an inherent mechanism for handling such scenarios.
>
> Case 2 connects to our new experiments and the reviewer's question 3:
>
> ### Second Generation of SENSEI Annotations
>
> > Since the distilled reward function is fixed during world model training, what if the agent discovers behaviors or states absent from the initial self-supervised data?
>
> Great point. If an agent encounters entirely novel states, the semantic reward may contain mostly noise, as it’s out of distribution for the trained VLM-Motif. Here, exploration relies on epistemic uncertainty rewards ($r^{dis}$), similar to Plan2Explore. In such cases, we can refine our notion of interestingness with new data. We adopt a generational perspective: run SENSEI for a first generation, then perform a second round of VLM annotations to distill new semantic rewards based on newly explored regions.
>
> We tested this in Pokémon Red, comparing SENSEI-General to Plan2Explore (see Reviewer 7mHN response for environment details and results). The initial SENSEI dataset contains 100K pairs from a Plan2Explore run (500K steps), reaching at most Viridian Forest. SENSEI reaches the frontier of Viridian Forest more often, occasionally breaking into Pewter City, where the first gym is located. However, the first-generation VLM-Motif has never seen this gym and thus relies on information gain to explore beyond Viridian Forest. After 750K steps, we sample 50K new pairs from a SENSEI run and re-annotate them, forming a refined second-generation semantic reward function ($R_\psi$) now informed by previously unseen maps.
>
> Qualitative results (Figure E2) on our rebuttal webpage [https://sites.google.com/view/sensei-icml-rebuttal](https://sites.google.com/view/sensei-icml-rebuttal) show that second-generation annotations correctly emphasize Pokémon Gyms—unlike the initial generation. This is essential for the agent to further explore the gym and beat the gym leader.
>
> We hope this motivates extending SENSEI to complex domains where new behaviors are unlocked through generations.
>
> ### Semantic Rewards and Evolving Experience
>
> > What happens if what’s considered interesting changes over time?
>
> While our semantic reward $r^{sem}$ is independent of agent experience, we assume a VLM inherently ranks more complex tasks (e.g., stacking) higher than simpler ones (e.g., pushing). However, SENSEI also incorporates epistemic uncertainty ($r^{dis}$) into its exploration reward ($r^{expl}$, Eq. 7), so the agent engages in interesting and yet novel behaviors. As exploration progresses, $r^{dis}$ naturally decreases as the world model improves (Eq. 6).
>
> Consider a case where the VLM assigns equal interestingness to pushing and stacking. Initially, both are uncertain, yielding similar $r^{expl}$. As the agent masters pushing, $r^{dis}$ diminishes, lowering $r^{expl}$ for that skill. The agent then prioritizes stacking—aligning exploration with increasingly complex tasks.
>
> ### Minor Comments
>
> **Human evaluations in Robodesk**
>
> > In the Robodesk evaluation, it would be useful to include a human evaluation of behaviors to support the claim that “our semantic exploration reward leads to more meaningful behavior than purely epistemic uncertainty-based exploration.”
>
> To avoid overstating, we revise the sentence to: “On average, our semantic reward leads to more object interactions than purely epistemic uncertainty-based exploration.”
>
> > Why is SENSEI-General not shown in MiniHack?
>
> The MiniHack prompt is already highly general. We describe it as a “rogue-like game” with a generic description and note the egocentric view, without explicitly mentioning doors, chests. The key is only mentioned if it is part of the agent's inventory (Supp C3.3). This differs from Robodesk, where we specify relevant objects. Given this broad framing, we don’t expect SENSEI-General to yield significantly different results.
>
> We appreciate the reviewer’s insightful questions and welcome any further discussion.

---

### Official Review · Reviewer_7mHN · 2025-03-14

**Overall Recommendation:** 3

**Summary:**

The paper proposes incorporating human priors into RL exploration to encourage policies to internalize a model of *interestingness*. This is done by first annotating pairs of frames for interestingness using VLMs (which, owing to training on internet-scale human data, has incorporated these priors). This is then distilled into a reward model, which in turn is distilled into the world model of a model-based Reinforcement Learning policy. The authors additionally use an epistemic uncertainty-based reward, incentivizing the agent to try new behaviors in interesting states.

**Claims And Evidence:**

While the experimental results are interesting, in my opinion, results on just two environments (Minihack, Robodesk) fall short of concretely establishing the claims.

**Essential References Not Discussed:**

N/A

**Experimental Designs Or Analyses:**

I did not find any major issues with the experimental designs.

**Methods And Evaluation Criteria:**

Yes

**Other Comments Or Suggestions:**

N/A

**Other Strengths And Weaknesses:**

While the paper proposes a novel take on intrinsic motivation in terms of human priors, the major weakness is the limited experimental analysis. I would like to see this framework extended to at least one other environment.

**Questions For Authors:**

* It appears that the two-stage process of transferring the VLM's interestingness analysis into the agent's world model -- via a distilled reward model -- may lead to significant information loss. At this stage, the approach seems like a proxy of a proxy for human interestingness, making it quite indirect and prone to potential failure points. Did the authors explore simpler alternatives?

**Relation To Broader Scientific Literature:**

Intrinsic Motivation and Reward Shaping quite important in the field of Reinforcement Learning.

**Theoretical Claims:**

N/A

---

> ### Author Rebuttal · Authors · 2025-04-01
>
> Thank you for reviewing our paper and highlighting that our research direction is important and acknowledging that our experimental results are interesting.
>
> ### New Environment: Pokémon Red
>
> As per your suggestion, we now apply SENSEI to another environment: the classic Game Boy game Pokémon Red. This is a challenging environment due to its (1) vast, interconnected world and (2) complex battle system requiring semantic knowledge (e.g. Water beats Rock). Strong exploration is essential for progressing toward the main goal: becoming the Pokémon Champion by defeating Gym Leaders. We use the PokeGym v0.1.1 setup [1], based on [2], with:
>
> - Observations: Only the raw 64×64 game screen, unlike previous RL applications which additionally used internal game states or memory [2,3].
> - Actions: A 6-button Game Boy action space (Left, Right, Up, Down, A, B).
> - Rewards: Event-based rewards following [1].
> - Episode Length: Episode length is increased with environment steps: 1k until 250k steps, 2k during 250–500k, 4k during 500–750k, etc. We take actions every 1.5seconds of gameplay.
> - Game Start: We skip the tutorial, until the agent can move freely and catch Pokémon, starting after receiving the Pokédex. We use the same checkpoint as [2], beginning with a level 6 Squirtle.
>
> We compare SENSEI-General to Plan2Explore. Both receive extrinsic task-based rewards alongside self-generated exploration rewards. SENSEI's initial dataset is collected using 500k steps of Plan2Explore (100K pairs). We use a general prompt:
>
> 'Your task is to help me play the Game Boy game Pokémon Red. I just obtained my starter Pokémon, Squirtle. My goal is to find and defeat the Gym Leader, Brock. What do I need to do, and which areas do I need to traverse to achieve this goal? Keep it very short.'
>
> We generate five different responses from GPT-4 and sample from them uniformly as context for image-based comparisons.
>
> The results are shown on our rebuttal webpage: https://sites.google.com/view/sensei-icml-rebuttal. We plot the histogram of the maximum map reached by the agent at each episode throughout the training run for SENSEI and Plan2Explore (5 seeds). Our results show that SENSEI progresses further into the game, and most notably manages to reach the first Gym in Pewter City, unlike Plan2Explore. Both SENSEI and Plan2Explore explore catching and training Pokémon, with SENSEI typically assembling a party of slightly higher levelled Pokémon than Plan2Explore. In order to succeed in the game you need to have a diverse team with high levels. This means achieving a high party level (higher counts on the right side of the histogram).As Pokémon unlocks new maps over time, we also explore using generations of SENSEI: refining the semantic reward using data from an earlier SENSEI run. See our response to Reviewer 9xtC for details.
> Thank you for your great suggestion of adding another environment. Let us know if you have further questions about our setup or results.
>
> ### Approximating human interestingness
>
> We assume “proxy-of-a-proxy” refers to our two-step distillation process of 1) first training motif from VLM annotations and 2) learning to predict semantic rewards in the world model. We understand the reviewer’s concern, but want to emphasize why both design choices are essential. We will explain this here in detail and add more emphasis on these aspects in our updated paper.
>
> 1. Distilling a semantic reward function $R_\psi$ from VLM annotations is necessary as it allows us to not deal with a large and slow VLM inside the fast RL loop. Additionally, directly querying VLMs can lead to noisy outputs, which can further derail the agent’s learning.
>
> 2. Learning a mapping from world model states to semantic rewards is a necessary design choice when working with latent state world models as we point out in the discussion and detail in Suppl. A. 3.
>
> World models such as the RSSM typically encode and predict dynamics fully in a self-learned latent state. Thus, for a world model to predict $r^{sem}_t$ at any point in time, we need a mapping from latent states to semantic rewards.
>
> This begs the question whether our mapping is the best way to do it. Another option would be to decode the latent state to images and use those as inputs for Motif. However, we believe this has several disadvantages: (1) Inefficiency—image decoding from latent states is expensive and slows down training. (2) Artifacts—decoded images can be blurry or unrealistic, causing a distribution shift and making Motif, trained on real environment images, encounter out-of-distribution errors.
> We hope this motivates our two-step distillation process. We thank the reviewer for allowing us to clarify this and update the paper to better emphasize this in the method section and discussion.
>
> [1]  Suarez (2024): https://pypi.org/project/pokegym/0.1.1/
>
> [2]  Whidden (2023), https://github.com/PWhiddy/PokemonRedExperiments,
>
> [3]    Pleines et al. (2025): Pokemon Red via Reinforcement Learning

---

> > ### Comment · Reviewer_7mHN · 2025-04-03
> >
> > Dear Authors,
> >
> > Thank you for the new experiments and detailed explanations. About ```Approximating human interestingness```, could you further elaborate on the following:
> >
> > 1. Distilling a semantic reward function from VLM: How does the size of the reward function module compare to the VLM? Could a smaller VLM be used instead of the distillation process?
> > 2. More importantly, what are the potential pitfalls of the two-stage approximation process that you use?

---

> > > ### Author Response · Authors · 2025-04-04
> > >
> > > Thank you for the follow-up questions.
> > >
> > > ### 1. Size of the Reward Function vs. VLM
> > >
> > > Our reward model (VLM-Motif) is a lightweight CNN, using under 1M parameters (see Suppl. A2. for the architecture), and thus significantly more efficient than even small VLMs, which would significantly slow down training when queried at each environment step.
> > >
> > > But here we want to highlight another important factor justifying the overall need for distillation: Even if we were to use a VLM in the loop, we need a way to extract a scalar reward signal given an observation image as input. Directly using VLM outputs lacks a stable frame of reference: without comparisons, rewards are not consistent across states or episodes. In contrast, VLM-Motif learns a scalar reward function using standard techniques from RLHF, which enforce transitivity and consistency by training on ranked observation pairs.
> > >
> > > ### 2. Pitfalls of the Two-Stage Approximation
> > >
> > > We see two potential pitfalls:
> > >
> > > (1) Reward smoothing — VLM annotations can be noisy, and both VLM-Motif and the reward head help smooth this signal. While this is often beneficial for learning stability, it can be risky when visually similar states should receive vastly different rewards.
> > >
> > > (2) Model capacity — VLM-Motif and the reward head must have sufficient expressiveness. If they are too small, they may underfit and fail to capture important semantic distinctions from the VLM.
> > >
> > > In the applications we considered, we did not encounter these issues; both our lightweight VLM-Motif and the default DreamerV3 output heads were sufficient.
> > >
> > > We thank the reviewer for raising these thoughtful questions and we will include a discussion of the pitfalls of our two-stage approximation process in the discussion section of the final version of the paper.
> > >
> > > We would appreciate it if the reviewer would reconsider their score in light of our new experiments in the new environment Pokémon Red and our added explanations regarding our double-distillation process.

---

### Decision · Program_Chairs · 2025-05-01

**Decision:**

Accept (poster)

**Comment:**

This paper studies using VLMs to derive a signal of "interestingness" for accelerating RL. The reviewers and I generally agree that the experiments are satisfactory evaluations of the method, and the method is a somewhat-useful contribution to the growing research area of using foundation models as a source of learning or exploration signals. The author responses addressed some of the remaining methodological questions. However, there are still lingering questions around the motivation for e.g. reliance on pre-collected data to initialize, whether the extent of the novelty is sufficient to merit publication, etc. The paper will certainly be of some interest, but potentially within a narrow community.